

# Recent changes in pan-Antarctic surface snowmelt detected by AMSR-E and AMSR2

Lei Zheng[1], Chunxia Zhou[1], Tingjun Zhang[2], Qi Liang[1], Kang Wang[3]

[1]Chinese Antarctic Center of Surveying and Mapping, Wuhan University, Wuhan 430079, China
[2]Key Laboratory of Western China's Environmental Systems (Ministry of Education), College of Earth and Environmental Sciences, Lanzhou University, Lanzhou 730000, China
[3]Institute of Arctic and Alpine Research, University of Colorado Boulder, Boulder, Colorado, 80309, USA

*Correspondence to*: Chunxia Zhou (zhoucx@whu.edu.cn) and Tingjun Zhang (tjzhang@lzu.edu.cn)

**Abstract.** Surface snowmelt in the pan-Antarctic, including the Antarctic sea ice and ice sheet, is crucial to the mass and energy balance in polar regions and can serve as an indicator of climate change. We investigated the spatial and temporal variations of the surface snowmelt over the entire pan-Antarctic as a whole from 2002 to 2017 by using the passive microwave remote sensing data. The stable orbit and appropriate acquisition time of the Advanced Microwave Scanning Radiometer for the Earth Observing System (AMSR-E) and the Advanced Microwave Scanning Radiometer 2 (AMSR2) enable us to take full advantage of the daily brightness temperature (Tb) variations to detect the surface snowmelt events. In this study, diurnal amplitude variations of AMSR-E/2 vertically polarized 36.5 GHz Tb (DAV36V) were utilized to map the pan-Antarctic snowmelt because it is unaffected by the snow metamorphism. We validated the DAV36V method against the ground-based measurements and further improved the method over the marginal sea ice zone by excluding the effect of open water. Snowmelt detected by AMSR-E/2 data agreed well with that derived by ERA-Interim reanalysis, and much more extensive than that detected by the Special Sensor Microwave/Imager (SSM/I) data. On average, pan-Antarctic snowmelt began on 19 September, and lasted for 32 days. Annual mean melt extent on the Antarctic ice sheet (AIS) was only 9% of that on the Antarctic sea ice. Overall, the pan-Antarctic surface snowmelt showed a trend (at 95% confidence level) toward later melt onset (0.70 days yr[-1]) during the 2002-2017 period. Surface snowmelt was well correlated with atmospheric indices in some regions. Notably, the decreasing surface snowmelt on the AIS was very likely linked with the enhancing summer Southern Annular Mode.

## 1 Introduction

Surface snowmelt on sea ice and ice sheets has a great influence on the energy and mass exchange between snow surface and the atmosphere because wet snow has a lower albedo, thus absorbing more incoming solar radiation than dry snow (Steffen, 1995). Snowmelt leads to the increase of snow grains and the formation of melt ponds on sea ice and ice sheets, which in turn absorb more radiation and induce further snowmelt (Picard and Fily, 2006; Kuipers Munneke et al., 2012; Tanaka et al., 2016). Meltwater may fill in the ice crevasses on ice sheets and migrate to the ice-bedrock surface, which can provide the conditions



for ice shelves to break up (Scambos et al., 2000; van den Broeke, 2005) and induce the acceleration of ice flow (Zwally et al., 2002; Sundal et al., 2011). Therefore, the spatial and temporal dynamics of surface snowmelt on sea ice and ice sheets have a direct effect on the mass and energy balances in polar regions (Abdalati and Steffen, 1997; Anderson and Drobot, 2001; Drobot and Anderson, 2001b; Belchansky et al., 2004; Picard and Fily, 2006; Markus et al., 2009; Mortin et al., 2012; Luckman et al., 2014). The timing and extent of surface snowmelt are indicators of changes in polar climate (Intergovernmental Panel on Climate Change (IPCC), 2014), and thus potentially have regional and global climate implications.

In-situ observations of snowmelt are sparse over sea ice and ice sheets due to the unfavorable environment. Remote sensing techniques can provide timely data sets for the monitoring of melt events in polar regions. The dielectric constant of snow is a function of frequency, snow temperature, density, salinity, ice-particle, volumetric liquid water content and water shape inclusions (Hallikainen et al., 1986; Proksch et al., 2015). When a snowpack starts to melt, the most significant change in the electromagnetic properties is an abrupt increase in the dielectric constant, which increases absorption and reduces the penetration depth (Ashcraft and Long, 2006). The radiation characteristics of a wet snow mixture are likely to be dominated by the dispersion behavior of liquid water, even when liquid water is only one percent by volume (Hallikainen et al., 1986). Melt signals, therefore, can be detected via microwave radiometry by identifying the sharp changes in microwave brightness temperatures (Tb) (Serreze et al., 1993; Liu et al., 2005; Bliss et al., 2017).

Microwave radiometers can operate in all atmospheric conditions. Most spaceborne passive microwave instruments provide more than two daily passes in polar regions. Various algorithms, including the single- and multi-channel methods, have been used to detect snowmelt based on radiometers. Scanning Multi-channel Microwave Radiometer (SMMR) 18 GHz and 37 GHz Tb, Special Sensor Microwave/Imager (SSM/I) 19 GHz and 37 GHz Tb have been used to detect surface snowmelt on ice sheets when Tb is above a region-specific or user-defined constant depending on the local snow properties (Ridley, 1993; Zwally and Fiegles, 1994; Mote and Anderson, 1995; Tedesco, 2009). Tb received by satellites is also related to the ground physical temperature, cloud and atmospheric water vapor, which may contaminate melt signals in the Tb time series. Multi-channel methods were developed to minimize these interferences, e.g., using a gradient ratio or a cross-polarized gradient ratio (XGPR) between 19 GHz and 37 GHz to detect surface snowmelt on sea ice and ice sheets (Steffen et al., 1993; Abdalati and Steffen, 1995; Drobot and Anderson, 2001a; Belchansky et al., 2004; Fettweis et al., 2007; Markus et al., 2009; Liang et al., 2013). Snowmelt can also be recognized by using edge detection or wavelet transform-based technologies due to the abrupt changes in Tb values for the transitions in freeze-thaw cycles (Joshi et al., 2001; Liu et al., 2005).

Snow grain size increases after the refreezing of snow liquid water. As a result, dry snow Tb decreases during melt season due to the increase of volume scattering (Markus et al., 2009). Therefore, single-channel methods may fail to identify melt events because of the metamorphic snow structures. Ramage and Isacks (2002, 2003) introduced the SSM/I diurnal amplitude variations (DAV) of 37 GHz Tb to investigate the snowmelt timing on the Southeast Alaskan Icefields. The DAV algorithm is unaffected by the snow metamorphism in melt detection. This technique has been successfully applied to the ice sheets and was proved to be more sensitive to snow liquid water than the XGPR method (Tedesco, 2007; Zheng et al., 2018). Freeze-thaw cycles on the Antarctic sea ice were also successfully detected based on the SSM/I 37 GHz DAV (Willmes et al., 2006,



2009). Furthermore, Arndt et al. (2016) distinguished temporary snowmelt from continuous snowmelt on the Antarctic sea ice by combining DAV and the cross-polarized ratio of SSM/I Tb. In these studies, surface snowmelt patterns have not been described over the marginal sea ice zone, where the earliest sea ice retreat has been identified based on the passive microwave sea ice concentration (SIC) measurements (Stammerjohn et al., 2008). DAV may be not strong enough to be identified as melt
signals when open water with low Tb emerges in the first-year sea ice zone.

Most of these studies investigated surface snowmelt on sea ice and ice sheet based on SSM/I sensors. However, the SSM/I observations showed considerable variations in local acquisition time because of the orbit degradation (Picard and Fily, 2006). By contrast, the Advanced Microwave Scanning Radiometer for the Earth Observing System (AMSR-E) and the Advanced Microwave Scanning Radiometer 2 (AMSR2) operates in controlled-orbits so that local acquisition time show little temporal
variation (http://www.remss.com/support/crossing-times). In addition, AMSR-E/2 can monitor the Antarctic sea ice and ice sheet (referred to as pan-Antarctic) at more appropriate local acquisition time (Zheng et al., 2018). Taking 2002-2003 as an example, the local acquisition time of ascending and descending SSM/I Tb product south of 40° S were 19.17±0.44 and 5.45±0.45, while they were 14.16±0.20 and 0.88±0.20 for AMSR-E Tb product. Compared with SSM/I, AMSR-E/2 have more opportunities to detect melt events in the pan-Antarctic due to a warmer and a colder period for ascending and descending
passes and an expected higher DAV.

Unlike the Arctic sea ice and Greenland ice sheet, the pan-Antarctic surface snowmelt is relatively short-lived and patchy (Drinkwater and Liu, 2000; Picard et al., 2007), and has received much less attention. Contrary to the reduction in sea ice extent, thickness and duration in the Arctic, the Antarctic sea ice presented increasing trends in both extent and duration, especially in the Ross Sea (Comiso and Nishio, 2008; Hobbs et al., 2016). Most of the Antarctic sea ice is snow-covered, even
in summer (Brandt et al., 2005). Melt ponds are not often observed on the Antarctic sea ice (Ackley et al., 2008). Meltwater on the Antarctic ice sheet (AIS) always refreezes instantaneously. Though more than 25% of the AIS has experienced snowmelt since the 1980s, only 2% melts every year (Picard et al., 2007). Melt extent has been decreasing over the AIS since 1987 (Tedesco et al., 2007).

Strong interactions have been found between the sea ice and ice sheet melting conditions through atmospheric circulation
(Stroeve et al., 2017). Previous studies always investigated surface snowmelt on sea ice and ice sheet separately, which may result in uncertainties in the integrated study. It is worthwhile to estimate snowmelt over the pan-Antarctic based on a uniform approach. The overall objective of this study is to improve the understanding of surface snowmelt over the pan-Antarctic in three aspects: (1) to detect the pan-Antarctic surface snowmelt at stable and appropriate local acquisition time based on AMSR-E/2, (2) to improve the performance of DAV algorithm in marginal sea ice zone by excluding the effect of open water, and (3)
to estimate the pan-Antarctic surface snowmelt as a whole, and systematically describe the surface snowmelt patterns and changes from 2002 to 2017.



## 2 Data sets

### 2.1 Data from AMSR-E/2

As a modified version of the AMSR radiometer, AMSR-E radiometer launched aboard the NASA Earth Observing System (EOS) Aqua satellite on 4 May 2002. We obtained the 25 km AMSR-E/Aqua L2A global swath spatially-resampled 36 GHz

Tb from the National Snow and Ice Data Center (NSIDC, www.nsidc.org) (Knowles et al., 2006). AMSR2 aboard the Global Change Observation Mission-Water (GCOM-W1) satellite was launched on 18 May 2012 after AMSR-E ceased operations. As a successor of AMSR-E, AMSR2 almost shares the same satellite orbit and sensor parameters as AMSR-E, providing continuous measurements for the study of global climate change, energy balance and ecosystems. AMSR2 Tb at a spatial resolution of 25 km used in this study was provided by the Japan Aerospace Exploration Agency (JAXA,

http://suzaku.eorc.jaxa.jp/GCOM/). AMSR-E/2 obtained data over a 1450 km swath. Tb at 36.5 GHz in vertical polarization ($Tb_{V36}$) was used to estimate pan-Antarctic snowmelt extent and timing in this study.

### 2.2 Sea ice and atmospheric products

The Bootstrap SIC product was used to mask the sea ice in this study (Comiso, 2017). The daily SIC product in the south hemisphere with a spatial resolution of 25 km was obtained from the NSIDC (http://nsidc.org/data/nsidc-0079), University of

Colorado Boulder. A sea ice pixel was determined when SIC is above 15% (Meier and Stroeve, 2008), and only the pixels with SIC above 80% for at least 5 days were considered in melt detection (Markus et al., 2009).

     ERA-Interim reanalysis is a sequential data based on the atmospheric model and assimilation system, which produced by the European Centre for Medium-Range Weather Forecasts (ECMWF). Advancing forward in time using 12-hourly analysis cycles, the ERA-Interim reanalysis includes various surface parameters, describing weather, ocean and land-surface conditions

since 1979 (Dee et al., 2011). The air temperature (Tair) of gridded ERA-Interim reanalysis was used to assist with melt detection based on AMSR-E/2, as well as derive snowmelt directly in this study.

## 3 Methodology

### 3.1 Tb and liquid water content

According to the Rayleigh-Jeans approximation, Tb is a function of emissivity (ε) and the near-surface physical temperature

(Ts) of snow and ice (Zwally, 1977):

$$Tb = \varepsilon Ts \tag{1}$$

     Previous studies always emphasize the process that Tb increases when the snowpack starts to melt due to the increase of emissivity. However, Tb decreases after reaching a peak when the volume liquid water is about one percent (Kang et al., 2014). The processes can clearly be seen in the simulations (Fig. 1). We investigated the variations in $Tb_{V36}$ with increasing snow

liquid water based on the Microwave Emission Model of Multi-layered Snowpack (MEMLS) (Wiesmann and Mätzler, 1999).



In the simulations, snow temperature, density and depth in a homogeneous snowpack were set to 273.15 K, 350 kg m$^{-3}$, and 20 cm by referring to Brucker et al. (2010) and Kang et al. (2014). Snow-ice reflectivity at 36.5 GHz was set to 0.045 in vertical polarization according to Powell et al. (2006). The evolution of Tb with increasing liquid water can be divided into energy saturation and energy dampening phases (Kang et al., 2014). The initial increases of Tb are accompanied by the

amplification of ε until the liquid water reaches a certain level. The subsequent energy dampening phase is characterized by monotonically decreases of Tb, which are caused by the increase of interface reflectivity due to the amplified real part of the refractive index (Kang et al., 2014).

Daily Tb variations for slight snowmelt are large because of the opposite freeze/thaw state. During the melt seasons, snow grain size can increase to 2 mm when meltwater refreezes in the pore space (Winebrenner et al., 1994), resulting in a much

lower Tb in dry snow regime (Markus et al., 2009) (Fig 2). Consequently, significant daily $Tb_{V36}$ variations also exist in the transition from dry to wet snow regime during the heavy melt season, even when day-time $Tb_{V36}$ is in the energy dampening phase. Diurnal freeze-thaw cycles in snowpack are prevailing in polar regions (Hall et al., 2009; Willmes et al., 2009; van den Broeke et al., 2010b), especially for the melt onset with daytime snowmelt and overnight meltwater refreezing. Single-channel thresholding methods may miss the melt events when Tb decreases due to the associated snow metamorphism, while DAV

algorithm can recognize melt signals through the melt season. Moreover, the optimum acquisition time of AMSR-E/2 enables us to take full advantage of the DAV algorithm in melt detection.

### 3.2 Melt detection methods

Vertically polarized SSM/I 37 GHz DAV has been used in melt detection on the Greenland ice sheet and the Antarctic sea ice (Willmes et al., 2006, 2009; Tedesco, 2007; Arndt et al., 2016). Extensively summer daily freeze-thaw cycles on the Antarctic

sea-ice were found by SSM/I (Willmes et al., 2009). AMSR-E/2 have more opportunities to identify these transitions due to the more appropriate local acquisition time. AMSR-E/2 36.5 GHz DAV in vertical polarization (DAV36), were used to detect the pan-Antarctic snowmelt and calculated as follow:

$$DAV36 = |Tb_{V36A} - Tb_{V36D}| \tag{2}$$

where $Tb_{V36A}$ and $Tb_{V36D}$ are the $Tb_{V36}$ in the ascending and descending passes respectively. Willmes et al. (2009) determined

melt signals when the DAV of SSM/I 37 GHz Tb exceeds 10 K. The threshold has been validated through extensive field data on the Antarctic sea ice. We utilized the same threshold for melt detection based on AMSR-E/2 DAV36 considering a very close channel.

This method was also applied in the investigation of snow surface freeze/thaw cycles on the AIS. In-situ measurements at Zhongshan Station (69.37°S, 76.38°E) obtained from the National Climatic Data Center (NCDC, www.chinare.org.cn) were

used to validate the DAV36 algorithm (Fig. 2). $Tb_{V36A}$ and $Tb_{V36D}$ showed sharp increases at melt onset, while decreased during melt seasons with associated snow metamorphism. By contrast, positive maximum Tair agreed well with melt signals determined by satellites, with an overall accuracy of 0.93 and a Kappa coefficient of 0.79.





When a snowpack starts to melt, it appears as a blackbody at microwave wavelengths and Tb increases sharply (Markus et al., 2009), while open water exhibits relatively much lower Tb (Markus and Cavalieri, 1998). So Tb amplitudes may be not strong enough to be identified as melt signals for the first-year sea ice with plenty of open water. To eliminate the effect of open water in melt detection, $DAV36_{ice}$ (i.e., DAV36 contributed by the ice-covered portion) was applied in the Antarctic sea ice snowmelt detection. Regardless of the atmospheric effects, Tb of sea ice is comprised by the ice portion ($Tb_{ice}$) and open water portion ($Tb_{ow}$) (Markus and Cavalieri, 1998):

$$Tb = Tb_{ice}\ SIC + Tb_{ow}\ (1-SIC) \tag{3}$$

therefore, $DAV36_{ice}$ can be calculated as:

$$DAV36_{ice} = \left| \frac{Tb_{V36A} - Tb_{ow}(1-SIC)}{SIC} - \frac{Tb_{V36D} - Tb_{ow}(1-SIC)}{SIC} \right| = \frac{\left| Tb_{V36A} - Tb_{V36D} \right|}{SIC} \tag{4}$$

Fig. 3 shows the comparison of meteorological observations of a sea ice buoy in the Weddell Sea and the corresponding AMSR2 measurements. Snow buoy observations (Fig. 3a), including snow depth and Tair, were obtained from the Alfred Wegener Institute (AWI, http://www.meereisportal.de/en/seaicemonitoring/buoy-mapsdata/). The insert map in Fig. 3b illustrates the annual mean SIC and the route of the buoy from multi-year ice to first-year ice in the Weddell Sea. $Tb_{V36A}$ and $Tb_{V36D}$ showed great differences in the melt season. Sporadic melt events were detected before December 2014. Accompanied by a slight decrease of snow depth, DAV36 exceeded 10 K and Tair went above the freezing point after mid-December. $DAV36_{ice}$ was almost equal to DAV36 when SIC was above 90%, while much higher than the later when SIC dropped after February. DAV36 was below 10 K with Tair above the freeing point for many times (see the black arrows), while these melt signals were successfully recognized by $DAV36_{ice}$. The overall accuracy and Kappa coefficient between the positive maximum Tair and the melt signals derived by satellites were 0.79 and 0.51 based on the DAV36, and were 0.82 and 0.60 based on the $DAV36_{ice}$. $DAV36_{ice}$ performs better than DAV36 in the marginal ice zone by reducing the effect of open water.

In order to capture complete melt seasons, a melting year starts on 1 July and ends on 30 June of the next year. The missing observations were filled based on time-line interpolation. In order to eliminate erroneous microwave signals, a median filter with a window of 3×3 was applied to the satellite observations. Further, surface snow on sea ice and ice sheet is supposed to be frozen when the daily maximum of ERA-interim Tair is below -5°C. Snowmelt indices, including melt onset (first day of snowmelt), and melting days were calculated based on the above method. Melt freeze-up and duration were not included in this study because plenty of sea ice melts quickly in austral summer and does not emerge any more in the melting year. Antarctic sea ice cover has exhibited considerable regional and annual variations (Hobbs et al., 2016). To be consistent, melting day fraction (MDF) and melt extent fraction (MEF) was introduced to study the snowmelt variation:

$$MDF = \frac{Melting\ days}{Ice\ cover\ days}, \quad MEF = \frac{Melt\ extend}{Ice\ cover\ extend} \tag{5}$$

where ice cover was determined with SIC > 15% for sea ice, and the AIS is assumed to be ice-covered all the year.



The verification of snowmelt is difficult, especially in the pan-Antarctic where meltwater refreezes quickly and climatic data are sparse. Nonetheless, surface snowmelt is determined by the atmospheric conditions and agrees well with the Tair distribution pattern (Tedesco, 2007; Liang et al., 2013). Once Tair approaches or exceeds the freezing point, meltwater emerges in the snow packs (Willmes et al., 2006). Owing to the solar radiation penetration and absorption within the snow pack, subsurface snow temperature can be higher than the surface on the AIS (Brandt and Warren, 1993), and snowmelt may occur when Tair is below the freezing point (Koh and Jordan, 1995; Zhang et al., 1996, 2001). Arctic sea ice freeze/thaw states were successfully determined when Tair is above -1°C (Markus et al., 2009). In this study, snowmelt over the pan-Antarctic was also determined by ERA-interim reanalysis when the daily maximum Tair goes above -1°C. The ERA-derived snowmelt was used to compare with satellite observations.

## 4 Results

### 4.1 Snowmelt distribution

The study area was divided into six parts to investigate regional discrepancy of surface snowmelt on sea ice and ice sheet according to Parkinson and Cavalieri (2012) (Fig. 4), namely the Weddell Sea (WS, 60 °W to 20 °E), Indian Ocean (IO, 20 °E to 90 °), Pacific Ocean (PO, 90 °E to 160 °E), Ross Sea (RS, 160 °E to 130 °W), Bellingshausen Amundsen Sea (BAS, 130 °W to 60 °W), and the AIS.

Integrated pan-Antarctic annual snowmelt was generated based on AMSR-E/2 (Fig. 5a-c). On average, pan-Antarctic snowmelt began on 19 September (DOY 81), and lasted for 32 days during 2012-2017 (Table 1). In general, snowmelt shows significantly latitudinal zonality. Melt onset came later from the marginal sea ice to the inland of the AIS, from low-latitudes to high-latitudes, while MDF decreased in an opposite trend (Fig. 5a-c). Annual mean melt onset, melting days and MDF of the six regions were also analyzed (Table 1). In terms of the satellite observation, the earliest snowmelt occurred in BAS where melt seasons began on 10 August (DOY 41) and melted for 40 days. Snowmelt on the AIS, as expected, began the latest on 4 December (DOY 158) and lasted for only 26 days on average. Surface snow of the multi-year sea ice in WS and BAS can melt for more than 100 days (Fig. 5b). MDF can reach 30% for the marginal sea ice of RS and WS (Fig. 5c). Snowmelt at high-latitudes was variable. For example, surface snowmelt on the Ross Ice Shelf extended to the inland area and even reached the Transantarctic Mountains in 2004-2005, but it was almost totally frozen during 2009-2011 (not shown).

Snowmelt onset derived by AMSR-E/2 and ERA agreed well with each other (Table 1 and Fig. 5), especially for the marginal sea ice. On average, melt onset derived by satellites was 6 days earlier than that detected by ERA. Local discrepancies existed in the near-shore sea ice where ERA detected earlier snowmelt onset (Fig. g). ERA recognized more melt events in WS (12d), BAS (16d), and RS (17d) where intense surface snowmelt was found (Fig. h). Satellite-based MDF for the marginal sea ice is lower than that derived by ERA (Fig. 5i). With the exception of the Antarctic Peninsula where the melt season can last for more than three months, AIS melt timing detected by the two methods were consistent with each other (Fig. 5k-o).




Seasonal evolution of snowmelt in different regions was examined by the normalized histograms of annual mean melt onset, melting days and MDF (Fig. 6). Notable differences existed between the temporal distribution of sea ice and ice sheet melt patterns, which can be seen in the two peaks for the pan-Antarctic melt onset histograms (Fig. 6a). Sea ice melt onset concentrated in mid-July for BAS, and in early August for IO. By contrast, the frequency of AIS melt onset did not reach the

peak until early January (Fig. 6a). Pan-Antarctic annual mean melting days and MDF were 32 days and 14% respectively (Table 1), while melting days histogram indicates a large number of transient snowmelt with only a few melt events, especially for the AIS (Fig. 6b). About 30% of the AIS experienced snowmelt over 2002-2017, however, about 67% of the AIS melt area melted for no more than 5 days. In general, melting days seldom exceeded 45 d, with the exception of the BAS where plenty of surface snow can melt for more than two months. The annual mean MDF in the BAS was 26% (Table 1), while MDF in

most of the AIS kept below 5% (Fig. 6c).

Daily melt extent on the Antarctic sea ice and AIS was calculated and presented in Fig. 7a. Annual mean melt extent on the AIS ($0.18 \times 10^6$ km$^2$) was only 9% of that on the Antarctic sea ice ($1.92 \times 10^6$ km$^2$). The AIS was almost totally frozen in winter (JJA) when plenty of first-year ice still melted. Sea ice extent decreased in October accompanied by the increasing sea ice melt extent. AIS melt extent began to extend about two months later. Sea ice melt extent reached the peak in mid-December with a

mean maximum of $7.01 \times 10^6$ km$^2$, while the peak of AIS melt extent appeared in mid-January with a mean maximum of $1.15 \times 10^6$ km$^2$. Sea ice melt extent peaked earlier due to the simultaneous decreasing sea ice extent. In mid-January, most of the Antarctic sea ice experienced surface snowmelt with a mean maximum MEF of 84%. The AIS MEF was much lower, and also reached the maximum (8%) in mid-January (Fig. 7 b).

## 4.2 Trend analysis

Trends in surface melting conditions during 2002-2017 were analyzed. Linear trends in the melt indices were calculated based on the annual departures. As listed in Table 2, the pan-Antarctic snowmelt as a whole showed a significant trend (at 95% confidence level) toward later melt onset (0.7 days yr$^{-1}$), especially in the RS, BAS and PO. Melt onset came significantly later (at 99% confidence level) in the RS (1.65 days yr$^{-1}$). Meanwhile, melting days and MDF in RS was also significantly decreased (at 95% confidence level). The largest trend in melt onset (2.13 days yr$^{-1}$) was observed in BAS. Though melt onset in PO

came significantly later (at 95% confidence level) with a rate of 1.44 days yr$^{-1}$, melting days and MDF have both increased in this period. The changes in melt indices in WS and IO were small and not significant. The AIS showed a negative trend in surface snowmelt with a later melt onset (0.79 days yr$^{-1}$), and slightly decreasing melting days and MDF.

The inter-annual departures of melt onset, melting days and MDF are presented for the pan-Antarctic, the AIS and the RS where the most significant changes in melting conditions were found (Fig. 8). Although the 2010-2011 season showed the

earliest melt onset, and largest MDF, the maximum annual mean melting days were found in the 2009-2010 season. Consistent with the pan-Antarctic, RS and AIS presented negative trends in surface snowmelt, which were indicated by all the melt indices. The earliest melt onset in RS were found in 2004-2005 when almost the entire Ross Ice Shelf also experienced snowmelt. All the AIS snowmelt indices indicated the weakest melt season in 2014-2015.



The considerable decrease of surface snowmelt in the RS can also be clearly seen in Fig. 9, which illustrates the trends in the melt indices during 2002-2017. Most of the pan-Antarctic showed a later melt onset, especially in RS and BAS where melting days and MDF also presented remarkable negative trends. Surface snow on the East Antarctic sea ice, especially in PO, also presented trends toward later melt onset; however, melting days and MDF in these regions have increased. Surface

snowmelt onset appeared earlier on the marginal sea ice in WS where melting days and MDF also increased. Many low-lying ice shelves in the AIS, such as the Larsen C Ice Shelf in the Antarctic Peninsula and the Abbot Ice Shelf in Marie Byrd Land, presented trends toward decreasing melting days and MDF.

## 5 Discussion

### 5.1 Comparisons

ERA detected earlier melt than AMSR-E/2 data (Fig. 5 and Table 1). This is because it takes time to produce liquid water when snow temperature approaches the melting point (Markus et al., 2009). Daily Tb variation is limited when liquid water does not refreeze or snowpack is still melting in the warm nights during heavy melt seasons (Willmes et al., 2009; Semmens and Ramage, 2014). As a result, ERA recognized more melt events for the regions where heavy snowmelt was found and the DAV algorithm fails to work, such as the WS, BAS, RS and the Larsen C ice shelf.

Willmes et al. (2009) established the SSM/I DAV algorithm to study the Antarctic sea ice surface snowmelt (hereafter W09). A remarkably later snowmelt onset was detected by W09 during 2002-2008 when compared with results derived from AMSR-E and ERA data in this study, especially for the marginal sea ice (Fig. 10). There are several reasons for the significant difference in snowmelt detection on sea ice by using AMSR-E and SSM/I data. First, W09 only studied surface snowmelt on sea ice after 1 October, in this study, we started from 1 July to 30 June in the next year. Second, daily Tb variations contributed

by snowmelt on sea ice portion were extracted by the $DAV36_{ice}$ algorithm (Eq. 4), snowmelt signals were amplified by reducing the effect of open water so that more melt events can be recognized (Fig. 3). Third, compared with SSM/I, AMSR-E operated in a stable orbit and observed the pan-Antarctic with more appropriate local acquisition time, and hence had more opportunities to identify melt events.

Considering snowmelt can be biased by various SMMR and SSM/I acquisition time, Picard and Fily (2006) retrieved

surface snowmelt of the AIS based on corrected 18-19 GHz Tb time series (hereafter PF06). AIS daily melt extent derived by AMSR-E/2 and ERA were consistent well with each other with $R^2$=0.93. PF06 daily melt extent also presented a high correlation with results from AMSR-E/2 ($R^2$=0.78). However, melt extent derived by PF06 was significantly smaller than that mapped by AMSR-E/2 (Fig. 11a). AMSR-E/2 and ERA daily mean melt extent were more than twice the melt extent mapped by PF06 from December to February (Fig. 11b). During the melt seasons, Tb may decrease due to the strong volume scattering

resulted from the snow metamorphism (Ramage and Isacks, 2002; Markus et al., 2009). Summer Tb can be even much lower than the winter observations (see in Fig. 2). PF06 determined snowmelt when SSM/I 19 GHz Tb exceeds the winter mean plus 2.5 times of standard deviation of winter Tb, therefore may underestimate surface snowmelt. Moreover, as explained in Section



3.1 and shown in Fig. 1, Tb decreases in the energy dampening phase during heavy snowmelt. Single-channel methods like PF06 may miss melt signals, while DAV algorithm is unaffected by the snow metamorphism and can detect snowmelt even in the energy dampening phase.

## 5.2 Uncertainties

There are several uncertainties in the pan-Antarctic snowmelt derived by AMSR-E/2 data. First, the DAV algorithm may fail to work when liquid water does not refreeze or snowpack is still melting in warm nights (Willmes et al., 2009; Semmens and Ramage, 2014). Fortunately, unlike the Arctic, surface snowmelt on the Antarctic sea ice is always patchy and relatively short-lived (Drinkwater and Liu, 2000). Second, although the DAV algorithm used in this study performs well when compared with ERA-interim reanalysis and meteorological observations, the optimal threshold may differ temporally and regionally with

varying snow properties. Arndt et al. (2016) utilized individual local thresholds with a median value of 6 K to detect snowmelt based on SSM/I DAV and recognized earlier melt onset than W09. Melt indices derived by the DAV algorithm showed considerable variations when applying different thresholds (from 6 to 14 K in Fig. 12). Varying the threshold applied to AMSR-E/2 DAV by ±4 K results in -12 days to 9 days departure for annual mean melt onset, and -6 days to 8 days departure for annual mean melting days. It is worth noticing that liquid water does not necessarily mean that a snowpack is melting because

it takes time for meltwater to refreeze (van den Broeke et al., 2010b). In addition, after the refreezing of surface snow, subsurface liquid water can still be detected by radiometer due to the penetrating capacity of microwave (Ashcraft and Long, 2006; Picard et al., 2007; Wang et al., 2016).

## 5.3 Response of the pan-Antarctic surface snowmelt to atmospheric indices

Snowmelt in the pan-Antarctic was found to be strongly associated with the atmospheric component of the El-Niño Southern

Oscillation (ENSO) and the Southern Annular Mode (SAM) (Turner, 2004; Tedesco and Monaghan, 2009; Oza et al., 2011; Meredith et al., 2017). In January 2016, the extensive surface snowmelt in west Antarctica was likely linked with sustained and strong advection of warm marine air due to the concurrent strong El Nino event (Nicolas et al., 2017). The weakly negative trend of surface temperature in Antarctica was consistent with the positive trends in the SAM during summer and autumn since the 1970s (Monaghan et al., 2008).

To study the response of the pan-Antarctic surface snowmelt to atmospheric conditions, we explored the relationship between melt indices and the Nino3.4 (Rayner et al., 2003), Southern Oscillation Index (SOI) (Ropelewski et al., 1987) and SAM (Marshall, 2003). No statistically significant correlations were found between these synoptic variables and the entire pan-Antarctic snowmelt. However, surface snowmelt was well correlated with these indices in some regions. An Anti-correlation was found between summer (DJF) SOI and MDF (R=-0.50, p<0.1) in BAS where strong decreases in sea-ice

concentration and duration were always linked with contemporary ENSO warm events (Kwok et al., 2002; Bromwich et al., 2004; Matear et al., 2015). MDF was negatively related to summer SAM for off-shore sea ice in BAS and RS where MDF decreased (Fig. 13a), this relationship was especially significant in the AIS (R=-0.88, P<0.01). High anti-correlations were



found between summer SAM and the annual mean MEF on the Antarctic sea ice and AIS (Fig. 13b). The decreasing annual mean melt extent on the AIS ($-0.37 \times 10^4$ km² yr⁻¹) during 2002-2017 was strongly associated with increasing summer SAM (R=-0.82, P<0.001). This phenomenon is in line with the decreasing surface snowmelt and the enhancing summer SAM in AIS since the 1970s (Marshall, 2003; Tedesco and Monaghan, 2009). Though AIS melt onset was also well correlated with

Nino3.4 (R=0.63, P<0.05), SAM was the principal driver of AIS near-surface temperature and snowmelt variability (Marshall, 2007; Tedesco and Monaghan, 2009). The positive SAM results in anomalously strong westerlies over the Southern Ocean and hence the reduction in poleward heat transport, leading to a subsequent atmospheric cooling in the Antarctic regions (Thompson and Solomon, 2002). The SAM is expected to have a continuous effect on the Antarctic climate and surface melting conditions in the next decades considering the projected ozone recovery (Thompson et al., 2011).

**6 Conclusions**

In this study, we investigated the pan-Antarctic surface melting conditions during 2012-2017, including melt onset, melting days, MDF and MEF, by using daily AMSR-E/2 Tb variations. Compared with SSM/I, the more stable orbit and more appropriate local acquisition time of AMSR-E/2 enable us to take full advantage of the DAV algorithm to investigate surface melt events. The performance of the DAV algorithm was improved in the marginal sea ice zone by excluding the effect of

open water. Though DAV algorithm may fail to recognize melt events when meltwater does not refreeze or snowpack even melts in the warm nights, snowmelt detected by AMSR-E/2 agreed well with that derived by ERA-interim reanalysis, and more extensive than that detected by SSM/I.

Pan-Antarctic snowmelt showed significantly latitudinal zonality. On average, the pan-Antarctic snowmelt began on 19 September (DOY 81). Sea ice in WS and BAS can melt for more than 100 days. Snowmelt on sea ice and ice sheet exhibited

great differences in temporal distribution patterns. Annual mean melt extent on the AIS was only 9% of that on the Antarctic sea ice. The pan-Antarctic showed a significant trend (at 95% confidence level) toward later melt onset (0.7 days yr⁻¹). Negative trends in snowmelt were found in RS, BAS, and AIS. The decreasing surface snowmelt in the AIS was very likely linked with the positive summer SAM trend during 2002-2017.

Though AMSR-E/2 observed the pan-Antarctic at the appropriate time for the snowmelt detection, they may underestimate

snowmelt because snowmelt can occur at any time and the refreezing is quasi-instantaneous. Other sources of microwave remote sensing data set, such as scatterometer and synthetic aperture radar, are expected to enrich the daily pan-Antarctic snowmelt observations in future works. Snowmelt derived by satellites can serve as input, as well as output validations for climate models.



## Acknowledgements

The authors would like to thank the National Snow and Ice Data Center (NSIDC) and the Japan Aerospace Exploration Agency (JAXA) for providing AMSR-E and AMSR2 data respectively. Sea ice concentration data was also obtained from NSIDC. European Centre for Medium-Range Weather Forecasts (ECMWF) is thanked for providing the ERA-interim reanalysis. The

numerical calculations in this paper have been done on the supercomputing system in the Supercomputing Center of Wuhan University. This research was funded by the National Natural Science Foundation of China (NSFC) (Grant no. 41376187, 41531069 and 41776200).

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



**Table 1. Annual mean and standard deviation of melt onset, melting days and MDF derived by AMSR-E/2 and ERA.**

| | Regions | WS | IO | PO | RS | BAS | AIS | All |
|---|---|---|---|---|---|---|---|---|
| **AMSR-E/2** | Melt onset | 19 Sep | 18 Sep | 30 Aug | 6 Sep | 10 Aug | 4 Dec | 19 Sep |
| | (DOY) | (81)±8 | (80)±7 | (61)±10 | (68)±8 | (41) ±14 | (158)±6 | (81)±5 |
| | Melting days (days) | 37±4 | 23±4 | 36±5 | 33±5 | 40±5 | 26±3 | 32±3 |
| | MDF (%) | 16±1 | 12±2 | 15±1 | 14±1 | 18±1 | 7±1 | 14±1 |
| **ERA** | Melt onset | 11 Sep | 16 Sep | 26 Aug | 24 Aug | 29 Jul | 3 Dec | 13 Sep |
| | (DOY) | (73)±10 | (78)±17 | (57)±14 | (55)±9 | (29) ±10 | (157)±5 | (75)±6 |
| | Melting days (days) | 49±3 | 26±5 | 41±5 | 50±3 | 56±6 | 26±2 | 42±2 |
| | MDF (%) | 21±1 | 14±2 | 16±2 | 22±2 | 26±2 | 7±1 | 18±1 |

**Table 2. Trends of snowmelt onset, melting days and MDF during 2002-2017. Trends with\*, \*\* and \*\*\* indicate statistical
10  significance at 90%, 95%, and 99% confidence levels, respectively.**

| Regions | WS | IO | PO | RS | BAS | AIS | All |
|---|---|---|---|---|---|---|---|
| Melt onset (days yr$^{-1}$) | -0.39 | 0.30 | 1.44* | 1.65*** | 2.13** | 0.79 | 0.70** |
| Melting days (days yr$^{-1}$) | 0.22 | 0.17 | 0.47 | -0.69** | -0.39 | -0.35 | -0.07 |
| MDF (% yr$^{-1}$) | 0.17 | 0.08 | 0.06 | -0.20** | -0.12 | -0.10 | 0.02 |





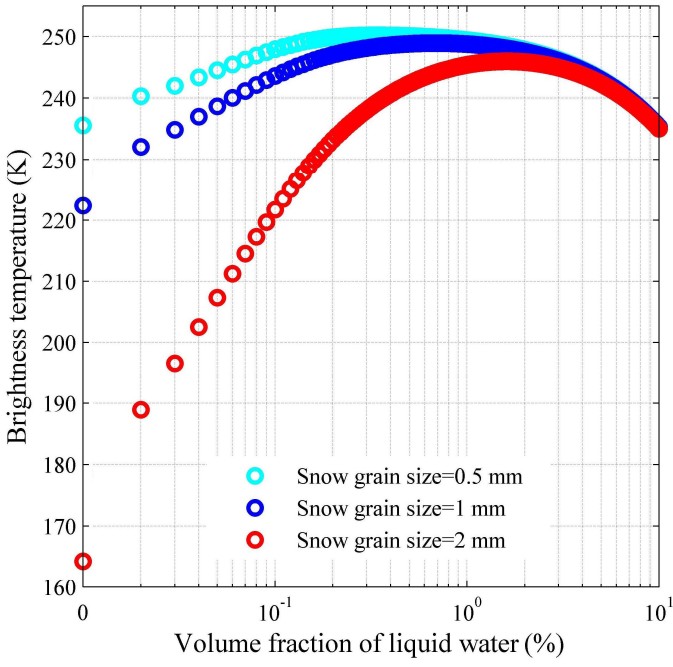

**Figure 1. Simulated Tb$_{V36}$ varying with liquid water when the snow grain size is 0.5 cm (cyan), 1 mm (blue) and 2 mm (red).**

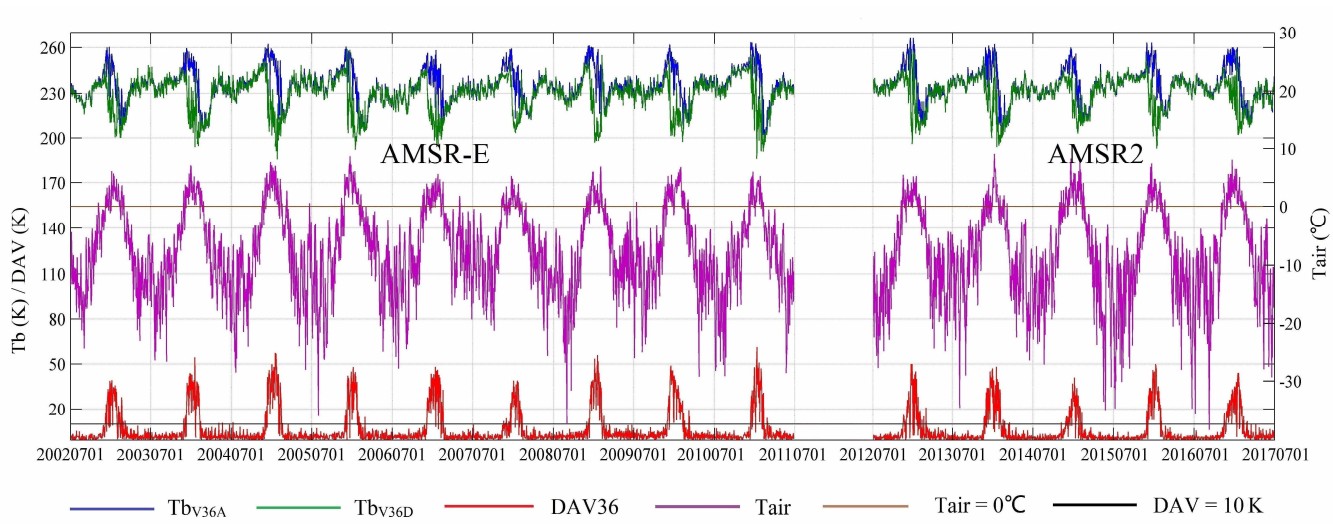

Figure 2. The comparison between Tair and satellite observations (AMSR-E from 2002 to 2011 and AMSR2 from 2012 to 2017) at Zhongshan Station, including daily maximum Tair (purple line), Tb$_{V36A}$ (dark blue line), Tb$_{V36D}$ (dark green line) and DAV36 (red line); Brown and black lines represent Tair = 0°C and DAV = 10 K.





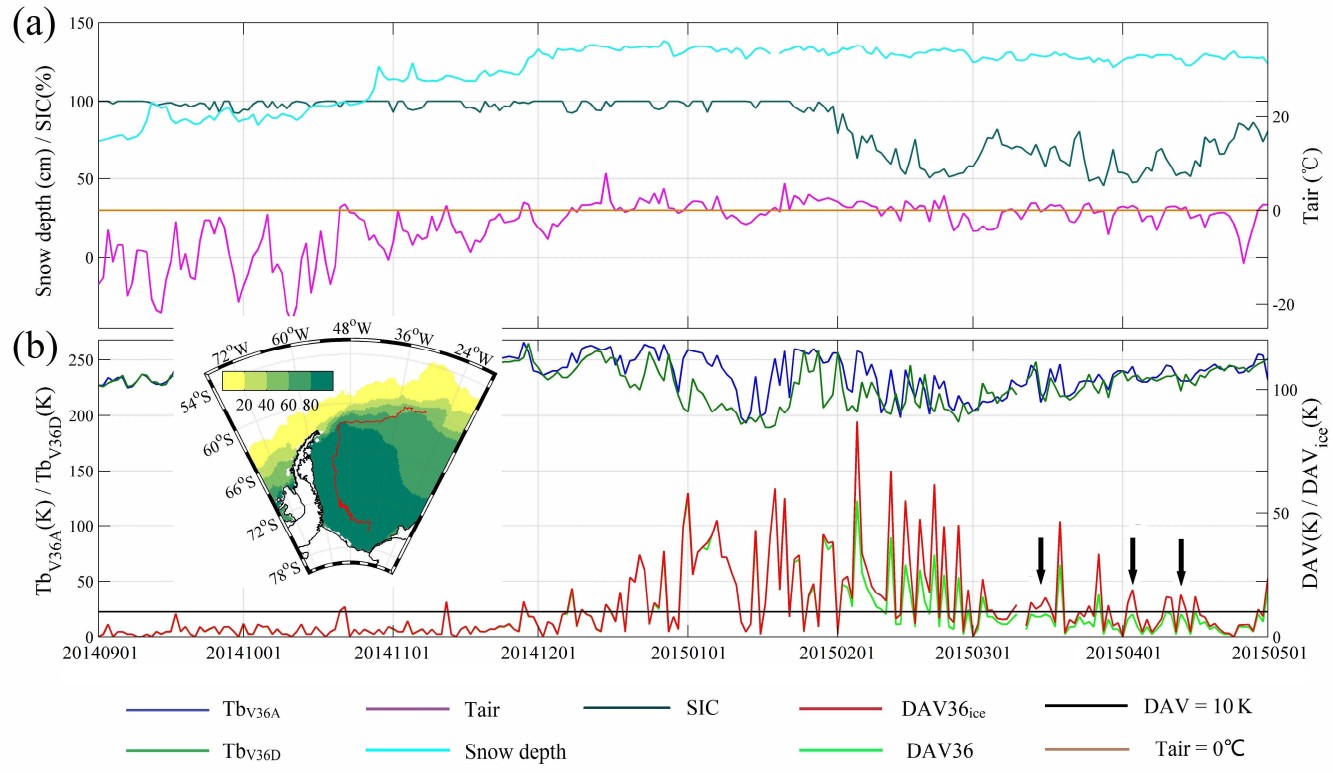

**Figure 3. Meteorological and satellite measurements along a sea ice buoy in the Weddell Sea from 1 September, 2014 to 1 May, 2015.**
**(a) Snow depth (cyan line), daily maximum Tair (pink line), and SIC (olive line); the brown line represents Tair = 0°C. (b) $Tb_{V36A}$**
**(dark blue line), $Tb_{V36D}$ (dark green line), DAV36 (light green line) and $DAV36_{ice}$ (red line); the black line represents DAV=10 K.**
**The inset map in (b) illustrates the annual mean SIC and the route of the buoy from multi-year ice to first-year ice. The black arrows**
**indicate the cases that melt events were recognized by $DAV36_{ice}$ rather than DAV36.**





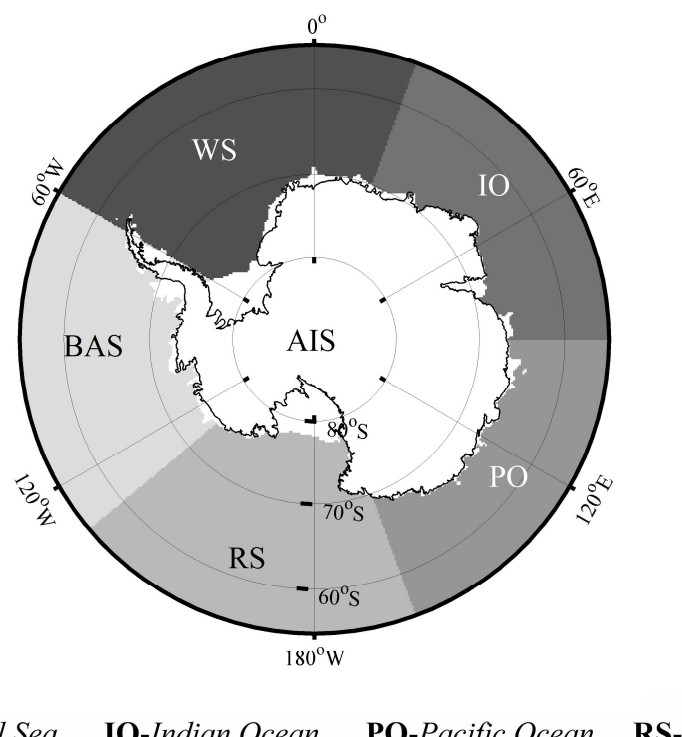

**WS**-*Weddell Sea*    **IO**-*Indian Ocean*    **PO**-*Pacific Ocean*    **RS**-*Ross Sea*

**BAS**-*Bellingshausen Amundsen Sea*    **AIS**-*Antarctic ice sheet*

**Figure 4. Map of the different regions across the Pan-Antarctic.**





**Figure 5. Annual mean melt onset, melting days and MDF derived by AMSR-E/2 (a-c) and ERA (d-f), also shown are the differences (AMSR-E/2 minus ERA) between the two observations (g-i).**



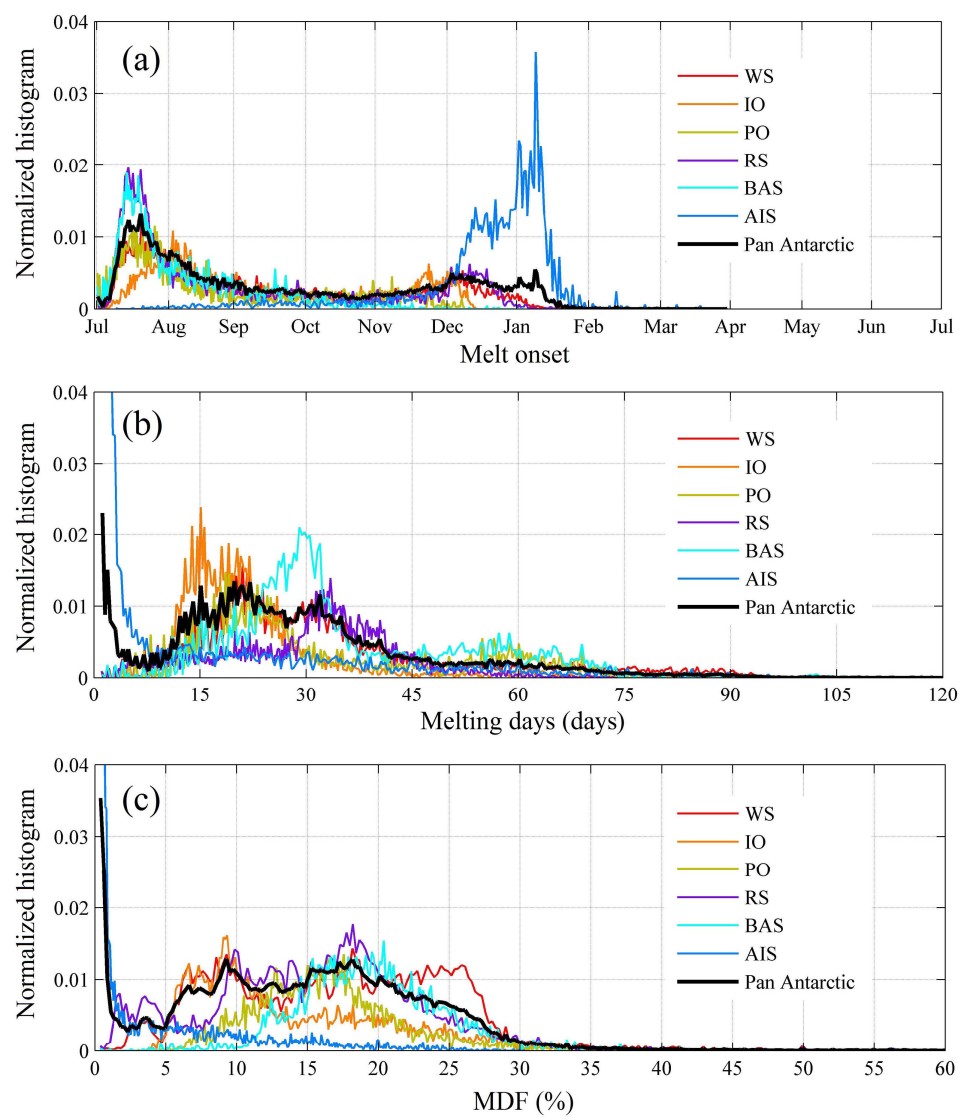

**Figure 6. Distributions of annual mean (a) melt onset, (b) melting days and (c) MDF derived by AMSR-E/2.**





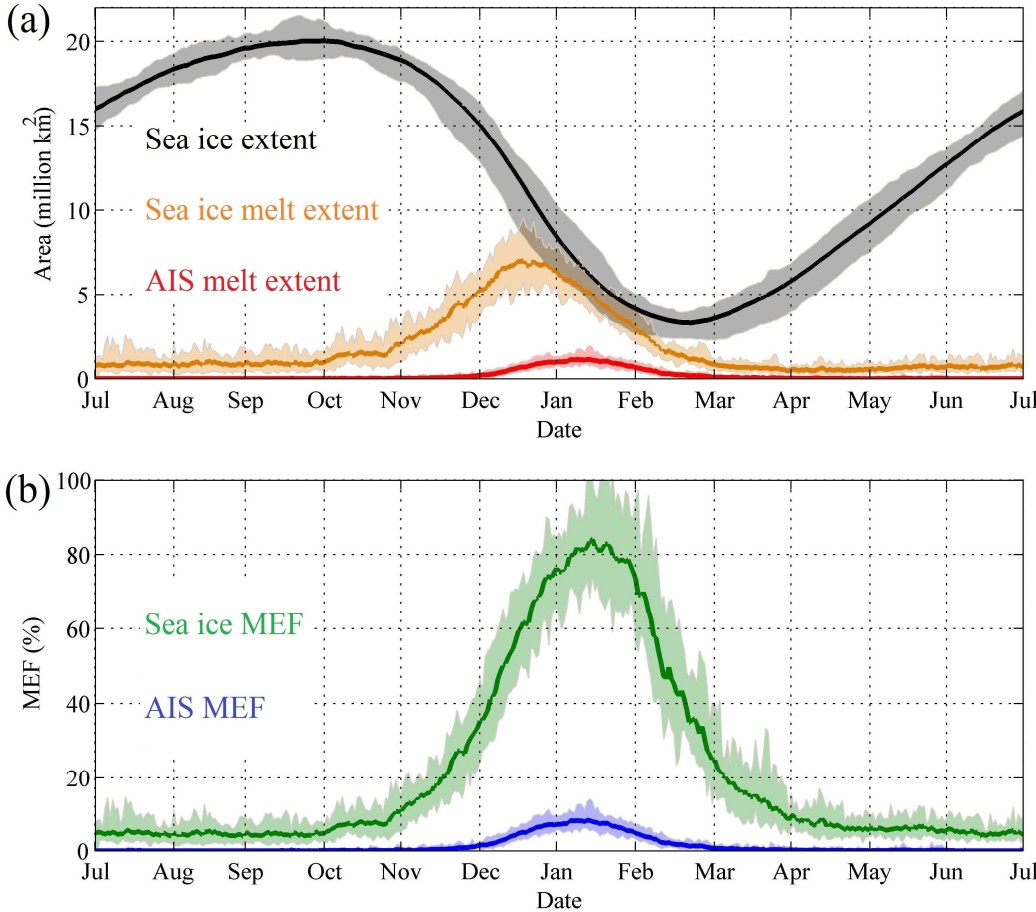

**Figure 7. Melt extent and MEF in the pan-Antarctic. (a) Daily mean sea ice extent (black line), sea ice melt extent (brown line) and AIS melt extent (red line); (b) daily mean sea ice MEF (green line) and AIS MEF (blue line); the corresponding shadows indicate daily maximums and minimums.**



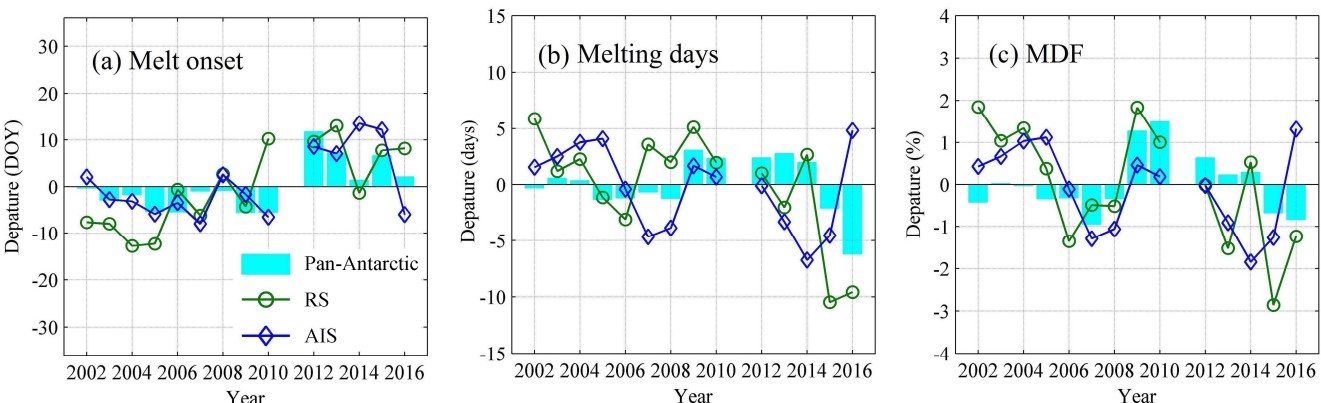

**Figure 8. Departure of annual mean (a) melt onset, (b) melting days and (c) MDF for the pan-Antarctic (cyan bar), RS (green line) and the AIS (blue line).**

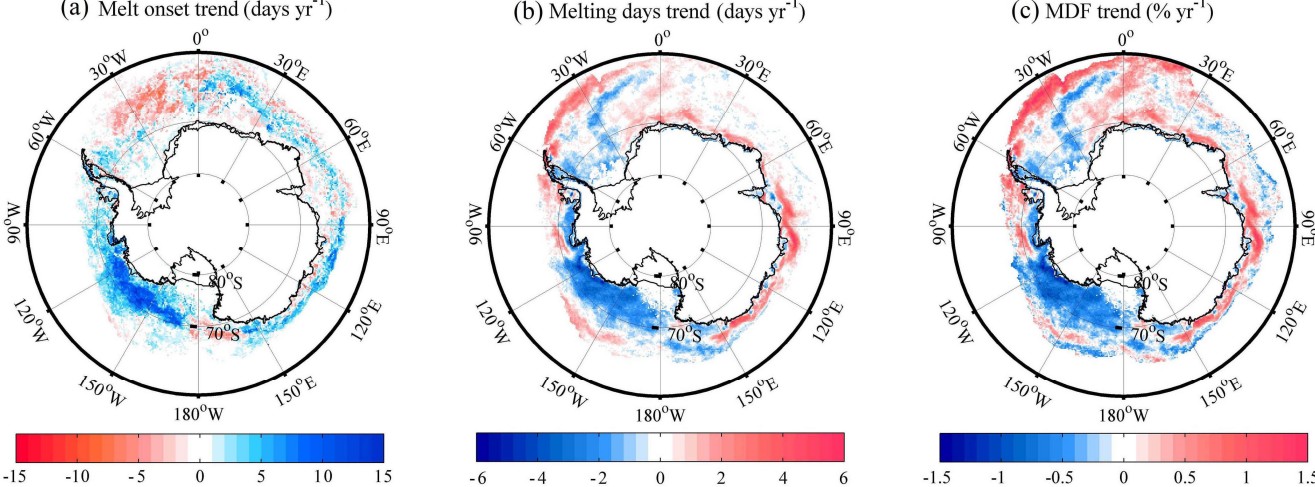

**Figure 9. Map shows the trend in (a) melt onset, (b) melting days and (c) MDF.**



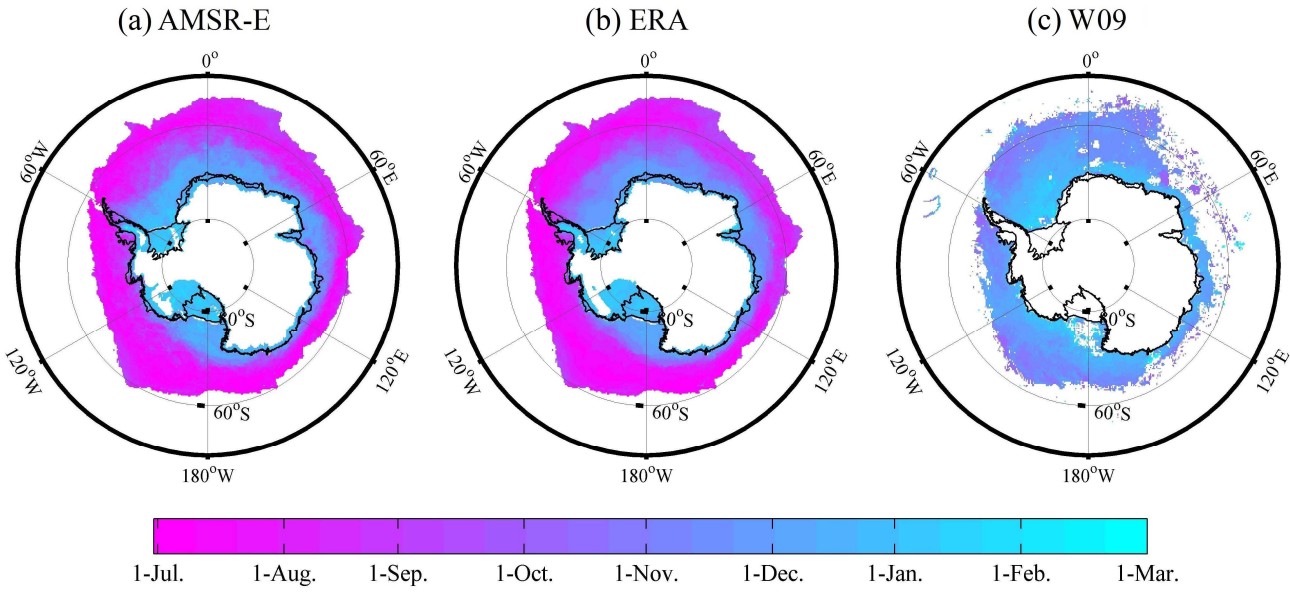

**Figure 10. Annual mean melt onset derived from (a) AMSR-E, (b) ERA (middle) and (c) W09 from 2002 to 2008.**

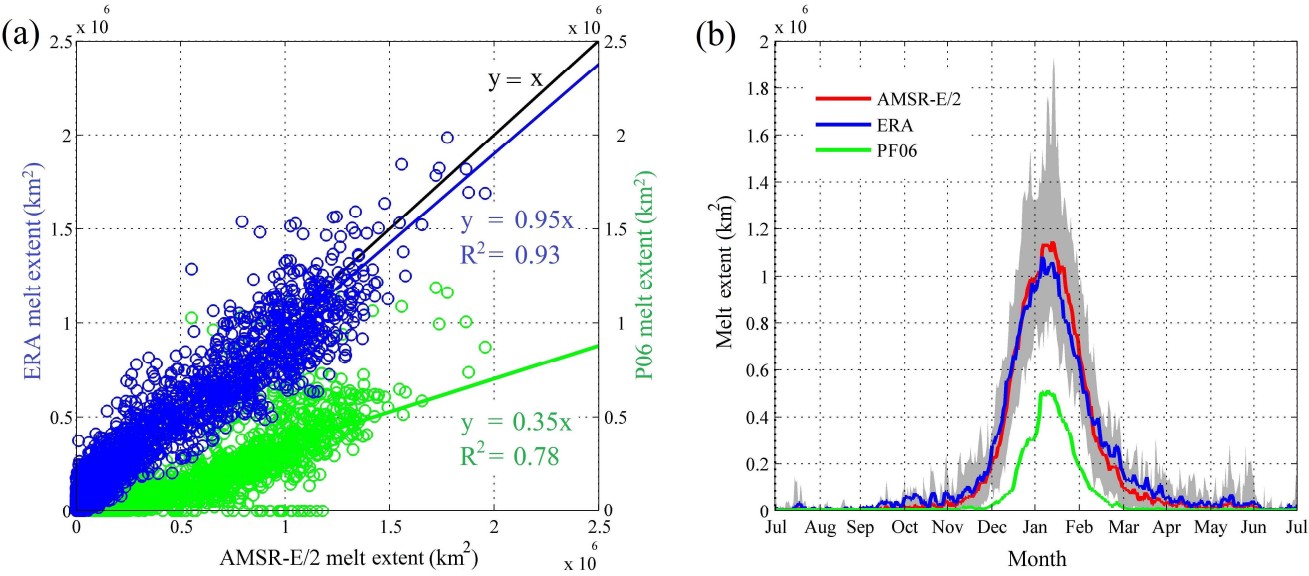

**Figure 11. Comparison of AIS melt extent derived by AMSR-E/2, ERA, and PF06 from 2002 to 2017. (a) Scatter plot of daily melt extent, blue circles indicate AMSR-E/2 vs. ERA, and green circles indicate AMSR-E/2 vs. PF06. (b) Daily mean melt extent derived by AMSR-E/2 (red line), ERA (blue line) and PF06 (green line), grey shadow indicates the daily maximum and minimum melt extent detected by AMSR-E/2.**



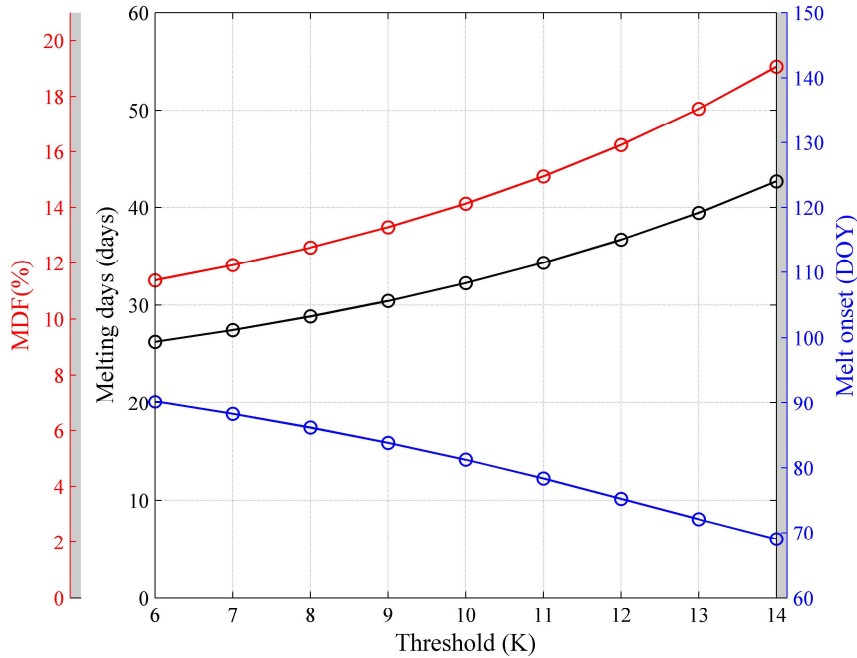

**Figure 12. Annual mean melt onset (blue dots), melting days (black dots) and MDF (red dots) with the threshold for AMSR-E/2 DAV36 varying from 6-14 K.**

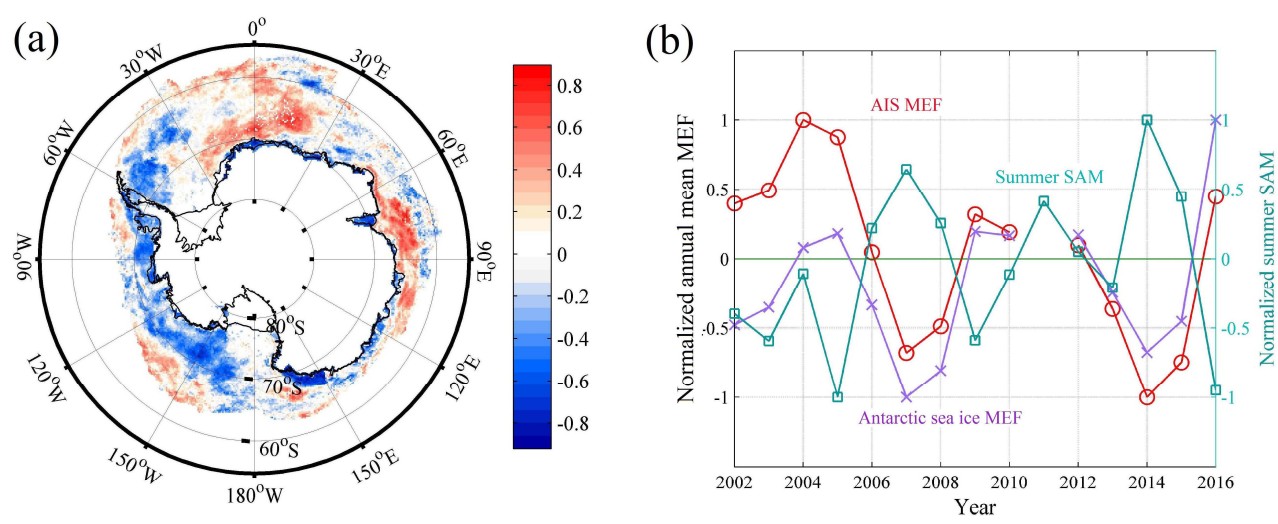

**Figure 13. Linkage between Pan-Antarctic snowmelt and summer SAM. (a) Correlation coefficient between MDF and summer SAM. (b) Comparison of normalized summer SAM (cyan line), normalized annual mean Antarctic sea ice MEF (purple line) and AIS MEF (red line).**