# Peer review of "Recent changes in pan-Antarctic surface snowmelt detected by AMSR-E and AMSR2"

_The Cryosphere, 2018_

## Referee Comment (RC1) · Anonymous Referee #1 · 3 Jun 2019

The presented paper is addressing the melt season in the Antarctic on the Antarctic ice shelves and the Antarctic sea ice cover. The research uses methods for the melt detection from AMSR-E and AMSR2 which are well established and the correction for ice concentration is promising idea to improve the melt onset detection. However, some of the analysis seems shallow and not well documented. Many information on how the results were obtained are missing.

General Comments:

1. The definition of melt is unclear in the manuscript. What exactly is supposed to be detected and discussed?

2. Some of the results regarding the melt onset and length are in line with other pa-

pers, however the melt onset of sea ice seems quite early in comparison to the cited references and other observations. Thus these early detected melt onset (July, August) need further physical investigation and should probably not directly interpreted as the real melt onset.

3. The optimal local acquisition time of AMSR-E compared to SSM/I repeatedly stated by the authors needs further explanation or investigation. It should be critically discussed whether other influences (maybe sun influences or instrument temperature) can alter the results (lead to too early snowmelt detection)

4. I'm surprised to see shelf and sea ice melt in the same way analyzed. They are so different in their nature and also physical properties that I would not even have expected that the same method would work adequately on both. For example there are brine and flooding effects in sea ice which are not present in the shelf ice. It should be more clearly stated in the Manuscript why it is useful or desired to combine the analysis.

5. The vast amount of references makes it very hard to find the the real sources for certain statements. This makes the manuscript appear cluttered and lacking a concrete direction and purpose.

Some specific Comments:

P1, L16: "DAV" should be directly introduced as TB_v difference of ascending and descending swaths either in the abstract or at the very first occurrence in the text

P2, L16 & L23: first statement is "passive microwave remote sensing works in all atmospheric conditions" and then "altered by clouds, atmosphere, . . .." what do you want to say here?

P3, L14: see General point 3.

P4, L15: SIC>15% was used and only SIC>80% was used in melt detection? Does this mean a pixel never exceeding 80% SIC is never melting? And pixels exceeding

80% SIC only later can only melt from this point on? I would expect that this gives you a negative bias in MDF (since it counts as frozen even in melting conditions).

P4, L20: please state the exact field of the ERA interim dataset used including timestep, are you using "Air temperature at 2m height" from the surface analysis? Also: in how far was the data used to "assist" with the AMSR-E/2 melt detection? Is this is described somewhere else in the text?

P4, L29-P5: It is unclear what the MEMLS simulation is for. In Kang et al. (2014), which you are citing four times in this paragraph, this is discussed in very detail. The variation of snow grain size is barely discussed in this paragraph and from what I got, never really picked up again in the manuscript. I would probably just remove the Fig. 1.

P5, L5-6: specify the interface you are talking about, probably the snow-air-interface

P5, L10-15: Since you employ the method, can you show that this signal is consistent characteristic for melt? For a longer constant melt under full sun illumination, there is probably not much difference between day and night wetness in the snow. Also in Fig. 3, under constant positive air temperatures, there is not constant DAV>10 which indicates that the melt indicator from DAV and the positive temperatures are not strictly connected.

P5, L15: see General point 3.

P5, L21: see General point 3.

P5. L31: I cannot see this in Figure 2. There are at least 3 years (2005, 2007,2009) where there is day in mid-winter with positive air temperature where DAV does not exceed the threshold nor shows any signal.

P5. L32: Accuracy and Kappa should be defined somewhere.

P6. L9 (Eq 4): This is only true under the assumption that SIC did not change within the

~12h from ascending to descending overflight. This should be mentioned. The method could be optimized in this regard by using the Tbs to retrieve the ice concentration in ascending and descending separately and then calculate the DAV with the aid of an open water tie point (which does not cancel out in case SICs are different for ascending and descending overflights)

P6. L20: accuracy and Kappa definition again

P6. L22: Why is a spatial median needed here, what are erroneous microwave signals?

P6. L25: "Melt freeze-up and duration. . .." - I don't understand what is meant here

P6. L29: extend -> extent

P7. L7: if below -5°C means frozen state (P6. L24) and above -1°C means melting, what state is there in between and how is that classified?

P7. L16: Discussion about Fig. 5, see also General 2.: the mid July melt onset around -60 to -65 latitude is quite surprising and needs discussion. Also the later melt onset in the more outer parts are interesting. Is it because there was no ice at the melt onset of the more southern regions and ice drifted there later so that melt occurs later in these regions? However, than the MDF should be even higher in these regions, probably close to 100%. I would also suggest not using the parula but a diverging colormap for the difference plot.

P7. L31: "Fig. 5k-o" -> "Fig. 5g-i"

P8. Discussion about Fig. 6: I would suggest splitting the histograms to maybe a 7 by 3 plot to be able to discuss the particular regions better. Also bin size in the histograms is too small, i.e., the histograms are to noisy to comfortable read their data.

P8. L11-12: The comparison of the melt extent of AIS with the sea ice area makes no sense in my opinion. The AIS is a much smaller region. What is the purpose of this comparison? I suggest to remove Figure 7 completely. However, the melt extent

is quite small in the early months like July/August on sea ice which actually contradicts the early melt onset in Fig. 6. This also indicates that the early melt onset is probably just a random noise effect since it does not cover a large area apparently.

P8. L17: with "mean maximum MEF" you mean the "Mean anual Maximum MEF" right? should than be changed in the text.

P9. L2: I actually do not understand how the trends are calculated. Fig 9 indicates that you calculate the trends pixel based. One would expect that neighbouring pixel having similar melt onset dates (Fig 9a). If the shown pixel based trends have any significance also the spatial pattern should be coherent.

P11. L 18: I suspect the values and discussion to change in case you reconsidered the early snowmelt onset

---

## Referee Comment (RC2) · Anonymous Referee #2 · 25 Jul 2019

This study makes use of passive microwave data from AMSRE and AMSR2 to detect melt over the Antarctic Ice Sheet and sea ice regions using a diurnal difference in brightness temperature algorithm. Means and trends in melt onset, number of melt days, and melt day fractions from 2002-2017 are presented and compared with ERA estimates of surface melt based on air temperature, and SSMI melt indices. A method of improving melt detection in marginal sea ice is also presented and validated.

In general the paper is well written and of great interest with excellent figures but a few points need to be addressed. There are many instances where the use (or not) of the definite article is incorrect, I suggest a read through by a native English speaker to correct these.

Early on it should be made clear that satellite algorithms for melt retrieval detect either

the presence or absence of liquid water, or the diurnal transition between the two, rather than actual melting events.

Various products used in this study, Tb, Tair, SIC were undoubtedly supplied at different projections and swaths and resolutions. Please provide more detail on how these products were coregistered.

The validation described briefly P5, L28 does not give enough detail. What is the 'melt signal determined by satellite'? How are the accuracy and coefficients referred to calculated? Why is this agreement 'in contrast'? In contrast to what?

P1, L21. It does not make sense to compare snow melt extents of sea ice and ice sheets when they cover different areas in total. What is the point?

P1, L28. You mean snow melt leads to an increase in size of snow grains.

P1, L30. You confuse ice sheets on bedrock, with the hydrofracture on ice shelves which are floating. Separate the discussion of these two impacts.

P3, L13. It needs to be made clearer why DAV is more likely to detect melt with AM-SRE/2. Ie Explain why time of day (rather than period) of the overpasses is important.

P3, L21. 'Meltwater on the AIS always refreezes instantaneously'. Needs a reference. Also,in this case, it would never be detected.

P4, L7. Changes 'almost shares' to 'shares almost'.

P4, L17. This sentence does not make sense. Please rewrite. Which air temperature was used? 2 m?

P4, L29. Please move reference to Fig. 1 to later in this paragraph. You have not yet described the simulations.

P5, L8. Replace 'opposite' with 'contrasting'.

P5, L12. Replace 'prevailing' with 'prevalent'.

P5, L19. Replace 'extensively' with 'extensive'.

P7, L19. This sentence needs rewording 'MDF decreases in an opposite trend' suggests that MDF decreases going from high to low latitudes.

P6, L24. How is this definition of frozen based on ERA Tair used in the algorithm?

P7, L31. Figs. 5k-o? should be g-i?

P8, L12. Again, what is the point of comparing melt extents of sea ice and AIS?

P8, L16. It is not clear why the decreasing sea ice extent would lead to an increasing sea ice melt extent? This would only explain the delayed peak in sea ice MEF.

Fig. 9. You should only plot those pixels with a significant trend. Or also plot the p values.

P9, L14. You should discuss further implications of the failure of DAV when melt is continuous. This would presumably manifest as a decrease in Melt Days detected where melt temporal continuity became more prevalent. Might this also explain areas with very early melt onset such as in BAS but a surprisingly low number of melt days.

P10, L25. Please include these correlations between atmospheric indices and melt in a table.

---

## Author Comment (AC1) · 28 Aug 2019

We are grateful to the referee for his/her time dedicated to this manuscript and for the constructive comments, which were all taken into account in the revised manuscript.

Authors' response was uploaded as a PDF file in the supplements.

Please also find the revised manuscript (clean copy), the revised manuscript with changes highlighted, and the additional supplement to the manuscript in the supplements.

Best regards,

Lei Zheng (on behalf of the co-authors)

[Figure]

Please also note the supplement to this comment:
https://www.the-cryosphere-discuss.net/tc-2018-279/tc-2018-279-AC1-supplement.zip
* * *

---

## Author Response (AR1)

We are grateful to the editor and referees for their time dedicated to this manuscript and for the constructive comments, which were all taken into account in the revised manuscript. Below, we answer point-by-point all the comments. The comments are reproduced in blue and the authors' responses are provided in black. The responses were coded in **RXCY** format with **X** indicating the referee number and **Y** indicating the referee comment number. The underlined texts are our corresponding changes in the revised manuscript.

The main changes in the revised manuscript include:

- Continuous melt onset (the first day when snowmelt lasts for at least three consecutive days) was added to investigate the pan-Antarctic snowmelt dynamics.
- The introduction section was revised to describe the motivation clearly and concisely.
- The melt detection methods and the evaluation method were described in more detail.
- Uncertainties of the trends were added.
- The comparison between melt extent on the ice sheet and sea ice was removed.
- Figure 5-13 were redrawn, two figures were added as supplements.
- The manuscript was edited by a native English speaker.

**Author Response to Referee #1**

The presented paper is addressing the melt season in the Antarctic on the Antarctic ice shelves and the Antarctic sea ice cover. The research uses methods for the melt detection from AMSR-E and AMSR2 which are well established and the correction for ice concentration is promising idea to improve the melt onset detection. However, some of the analysis seems shallow and not well documented. Many information on how the results were obtained are missing.

General Comments:

1. The definition of melt is unclear in the manuscript. What exactly is supposed to be detected and discussed?

**R1C1:** Snowmelt detected by radiometers is actually the presence of snow liquid water (Zheng et al., 2019). We clarified this issue in the revised manuscript:

Therefore, snowmelt can be detected via microwave radiometry by identifying the sharp changes in microwave brightness temperatures (Tb) caused by the presence of snow liquid water (Serreze et al., 1993; Liu et al., 2005).

Although snow liquid water is produced by surface snowmelt, the presence of liquid water does not always indicate snowmelt. This is because liquid water may remain in snowpack after intense surface snowmelt. However, to be consistent with the previous studies (e.g., Abdalati and Steffen, 1995; Picard and Fily, 2006; Tedesco, 2009; Willmes et al., 2009; Arndt et al., 2016), "snowmelt" is still used in this study. We clarified this issue in Section 5.2 (Uncertainties):

> It should be noted that the presence of snow liquid water detected by AMSR-E/2 does not necessarily mean that the snowpack is melting because it takes time for meltwater to refreeze. In addition, after refreezing of surface snow, subsurface liquid water can still be detected by radiometer due to the penetrating capacity of microwaves (Ashcraft and Long, 2006).

2. Some of the results regarding the melt onset and length are in line with other papers, however the melt onset of sea ice seems quite early in comparison to the cited references and other observations. Thus these early detected melt onset (July, August) need further physical investigation and should probably not directly interpreted as the real melt onset.

**R1C2:** We thank the referee for this insightful comment.

- Melt onset investigated in this study (the first day that snowmelt is detected) is different from that (the first day when snowmelt lasts for at least three consecutive days) examined in Willmes et al. (2009). In previous studies, the former one was also defined as "early melt onset" (e.g., Semmens et al., 2013; Bliss et al., 2017), while the latter one was regarded as "continuous melt onset" (e.g., Markus et al., 2009) or "persistent melt onset" (e.g., Zheng et al., 2018).
- Early melt onset (EMO) always occurs much later than continuous snowmelt onset (CMO), especially on the first-year sea ice (Fig. R1). Surface melt on the Antarctic sea ice periodically occurs in winter (Massom et al., 2001). In August 2002, early melt events were observed on the first-year sea ice in the Ross Sea, while CMO did not occur until early November (Fig. R1c).

[Figure]

Fig. R1. Surface snowmelt detection on the Antarctic sea ice. (a) Pan-Antarctic ice cover days in 2002-2003, Point B and C show the locations of the pixels examined in (b) and (c). (b) and (c) show the comparisons of sea ice concentration (SIC), ERA-Interim Tair and satellite observations for a multi-year sea ice pixel (Point B) and a first-year sea ice pixel (Point C). $DAV37_{ice}$ and $DAV36_{ice}$ denote diurnal amplitude variations (DAV) of vertically polarized SSM/I 37 GHz Tb and AMSR-E 36.5 GHz Tb contributed by the ice portion, respectively.

- To compare with the results from Willmes et al. (2009) (hereafter W09), CMO was also included in the revised manuscript. Two different kinds of "melt onset" were investigated in this study:

  Considering the existence of both transient and persistent snowmelt in the pan-Antarctic, early melt onset (EMO, the first day when snowmelt is detected) and continuous melt onset (CMO, the first day when snowmelt lasts for at least three consecutive days) were investigated in this study. On average, CMO was about 53 days later than EMO. CMO derived from AMSR-E and W09 agreed well with each other at high latitudes during 2002-2008. However, AMSR-E found an earlier CMO on the marginal sea ice compared with the results from W09 (Fig. 10). The reasons for their differences were explained:

  First, W09 only studied surface snowmelt on sea ice after 1 October, while the melt season begins on 1 July in this study. Second, the $DAV36_{ice}$ algorithm can amplify snowmelt signals by reducing the effect of open water, so that more melt events can be recognized (Fig. 3). Third, compared

with SSM/I, AMSR-E operated in a stable orbit and observed the pan-Antarctic with more appropriate local acquisition time, and hence had more opportunities to identify melt events (Supplement Fig. 2).

[Figure]

Figure 10. Annual mean CMO derived from (a) AMSR-E and (b) W09 from 2002 to 2008.

3. The optimal local acquisition time of AMSR-E compared to SSM/I repeatedly stated by the authors needs further explanation or investigation. It should be critically discussed whether other influences (maybe sun influences or instrument temperature) can alter the results (lead to too early snowmelt detection)

**R1C3:** The comparison between SSM/I and AMSR-E Tb observations over dry snow zone suggest the effect of other influences is very limited.

- Dai and Che (2010) have compared the SSM/I and AMSR-E Tb observations, and concluded that the differences between AMSR-E vertically polarized 36.5 GHz Tb ($Tb_{V36}$) and SSM/I vertically polarized 37 GHz Tb ($Tb_{V37}$) were small.
- To further examine the interferences from cross-platform, we compared AMSR-E $Tb_{V36}$ and SSM/I $Tb_{V37}$ south of 85° S where surface snow is stable and never melts. R-square between AMSR-E $Tb_{V36}$ and SSM/I $Tb_{V37}$ were both 0.96 for ascending and descending passes during 2002-2003. Bias between the two measurements was only about 1 K. Bias between AMSR-E DAV36V and SSM/I DAV37V were less than 0.4 K. The effect of slight Tb offsets between

different sensors should not affect the melt detection based on temporal Tb variability (Markus et al., 2009).

[Figure]

Fig. R2. Comparison between AMSR-E $Tb_{V36}$ and SSM/I $Tb_{V37}$ during 2002-2003. (a), (b) and (c) show the comparisons for ascending passes, descending passes, and the DAV.

- We clarified this issue in the revised manuscript:

  We utilized the same threshold for melt detection based on AMSR-E/2 DAV36 considering the differences between AMSR-E 36 GHz Tb and SSM/I 37 GHz Tb are very small (Dai and Che, 2010). In the region south of 85° S where the surface snow is stable and never melts, the bias between the two measurements was only approximately 1 K during 2002-2003 (Supplement Fig. 1). Slight Tb offsets between different sensors should not affect the results when using temporal Tb variability in melt detection (Markus et al., 2009).

4. I'm surprised to see shelf and sea ice melt in the same way analyzed. They are so different in their nature and also physical properties that I would not even have expected that the same method would work adequately on both. For example there are brine and flooding effects in sea ice which are not present in the shelf ice. It should be more clearly stated in the Manuscript why it is useful or desired to combine the analysis.

**R1C4:** A uniform approach was applied in melt detection on both the sea ice and ice sheet for two reasons:

- First, snowmelt on the ice sheets was found to be correlated with that on the sea ice, but melt detection on sea ice and ice sheet was always conducted separately. This may result in uncertainties in the integrated study.

  Recent studies (e.g., Ballinger et al., 2013; Stroeve et al., 2017) found that ice sheet atmospheric pattern and snowmelt are linked with sea ice melting conditions through atmospheric circulation.

Earlier melt onset of the sea ice may have provided an additional source of warm, moist air over the adjacent ice sheet, leading to the earlier arrival of melt onset on the ice sheet. However, snowmelt over sea ice and ice sheet was always separately detected with different approaches. In Stroeve et al. (2017), sea ice melt onset was investigated based on Tb temporal variation and gradient ratio following Markus et al. (2009), while the ice sheet melt onset was determined based on a single-channel method following Mote (2007).

In the Antarctic, snowmelt on the West Antarctic and Antarctic Peninsula was also found to be linked with adjacent sea ice variations (Scott et al., 2018; Zheng et al., 2019). So it is worthwhile to generate integrated snowmelt over the pan-Antarctic.

- Second, the DAV method has been successfully applied in melt detection on both sea ice (Willmes et al., 2009) and ice sheet (Tedesco, 2007; Zheng et al., 2018). In addition, the thresholds used for melt detection on the Antarctic sea ice (10 K) and ice sheet (9 K) are very close. A threshold of 10 K works well in melt detection on both sea ice and ice sheet compared with the positive Tair observations (Figure 2&3).

Therefore, this study aims at generating integrated pan-Antarctic surface snowmelt based on the DAV method. We considerably revised the introduction section and clearly clarify the motivations in the last paragraph:

Strong interactions have been found between sea ice and ice sheet surface snowmelt through atmospheric circulation (Stroeve et al., 2017). Surface snowmelt dynamics in the West Antarctic and Antarctic peninsula have been found to be related with the sea ice variations in adjacent seas (Scott et al., 2018; Zheng et al., 2019). Previous studies have separately investigated surface snowmelt on sea ice and ice sheet, which may result in uncertainties in the integrated study. The DAV method has been successfully applied in snowmelt detection on both sea ice (Willmes et al., 2009) and ice sheet (Tedesco, 2007; Zheng et al., 2018). It is worthwhile to estimate snowmelt over the pan-Antarctic based on a uniform approach. The overall objective of this study is to improve the understanding of surface snowmelt over the pan-Antarctic based on the DAV method in three aspects: (1) to detect the pan-Antarctic surface snowmelt at the stable and appropriate local acquisition time based on AMSR-E/2, (2) to improve the performance of the DAV method in the marginal sea ice zone by excluding the effect of open water, and (3) to estimate the pan-Antarctic surface snowmelt as a whole and systematically describe the surface snowmelt patterns and changes from 2002 to 2017.

We acknowledge that brine and seawater flooding could affect the melt detection on sea ice, which was discussed in Section 5.2 (Uncertainties):

Second, although the DAV method used in this study performs well when compared with meteorological observations, the optimal threshold may differ temporally and regionally with varying snow properties. In addition, ice disintegrates, brine and flooding effects may play an important role in seasonal and even diurnal sea ice Tb variations, further complicating the story (Smith, 1998; Willmes et al., 2009).

5. The vast amount of references makes it very hard to find the the real sources for certain statements. This makes the manuscript appear cluttered and lacking a concrete direction and purpose.

**R1C5:** We thank the referee for pointing out this issue. We removed the less relevant references. The introduction was also considerably revised to make the motivations and directions clear.

Some specific Comments:

P1, L16: "DAV" should be directly introduced as TB_v difference of ascending and descending swaths either in the abstract or at the very first occurrence in the text

**R1C6:** DAV was directly introduced in both the abstract:

In this study, the difference between AMSR-E/2 ascending and descending 36.5 GHz Tb in vertical polarization (DAV36) was utilized to map the pan-Antarctic snowmelt because it is unaffected by the snow metamorphism.

and the text:

Ramage and Isacks (2002, 2003) introduced the SSM/I diurnal amplitude variations (DAV, i.e., the Tb difference between ascending and descending passes) in vertically polarized 37 GHz Tb to investigate the snowmelt timing on the Southeast Alaskan Icefields.

P2, L16 & L23: first statement is "passive microwave remote sensing works in all atmospheric conditions" and then "altered by clouds, atmosphere, …." what do you want to say here?

**R1C7:** Good catch. Although atmospheric effects are generally negligible in melt detection, they could potentially influence the melt signals and introduce errors (Abdalati and Steffen, 1995). Two avoid contradiction between the two statements, we revised the first sentence:

Microwave radiometers can operate regardless of illumination conditions and are insensitive to atmospheric conditions.

P3, L14: see General point 3.

**R1C8:** The comparisons between SSM/I and AMSR-E Tb observations over dry snow zone suggest the differences between the two measurements are very small. Please see **R1C3** for full details.

In addition, AMSR-E/2 can observe the pan-Antarctic snowmelt at more appropriate local acquisition time for two reasons:

- First, AMSR-E and AMSR2 operate in controlled-orbits measurements, and the crossing time for the two sensors are nearly the same. By contrast, crossing time differs between SSM/I sensors and also changes significantly over the years of operation due to orbit degradation (Picard and Fily, 2006) (Fig. R3). AMSR-E/2 measurements with a stable orbit are superior in the analyses of inter-annual snowmelt dynamics.

[Figure]

Figure R3. Ascending (solid lines) and descending (dash lines) equatorial crossing times for microwave sensors. The chart is adopted from Remote Sensing Systems (http://www.remss.com/support/crossing-times/).

- Second, the Antarctic diurnal melt area varies approximately as a sinusoid with the peak in the afternoon and the trough in the early morning (Picard and Fily, 2006). It is a great opportunity for us to make full use of the AMSR-E/2 data to detect surface snowmelt because the ascending and descending passes of AMSR-E/2 observed the pan-Antarctic in the afternoon (the warmest period) and at midnight (a cold period).

We rephrased this paragraph to make it clear:

Most of these studies investigated surface snowmelt on sea ice and ice sheets based on SSM/I sensors. However, SSM/I observations show considerable variations in local acquisition time

because of orbit degradation (Picard and Fily, 2006). By contrast, the Advanced Microwave Scanning Radiometer for the Earth Observing System (AMSR-E) and the Advanced Microwave Scanning Radiometer 2 (AMSR2) operate in controlled-orbits so that local acquisition time shows little temporal variation (http://www.remss.com/support/crossing-times). AMSR-E/2 measurements with a stable orbit are superior in the analyses of inter-annual snowmelt dynamics. Diurnal melt area in the Antarctic varies approximately as a sinusoid with the peak in the afternoon and the trough in the early morning (Picard and Fily, 2006). AMSR-E/2 can monitor the Antarctic sea ice and ice sheet (referred to as pan-Antarctic) surface snowmelt at the appropriate local acquisition time. Taking 2002-2003 as an example, the local acquisition time of ascending and descending SSM/I Tb products south of 40° S were 19.17±0.44 and 5.45±0.45, respectively, while these values were 14.16±0.20 and 0.88±0.20 for the AMSR-E Tb products. Compared with SSM/I, AMSR-E/2 have more opportunities to detect melt events in the pan-Antarctic due to warmer and colder periods for ascending and descending passes and an expected higher DAV.

P4, L15: SIC>15% was used and only SIC>80% was used in melt detection? Does this mean a pixel never exceeding 80% SIC is never melting? And pixels exceeding80% SIC only later can only melt from this point on? I would expect that this gives you a negative bias in MDF (since it counts as frozen even in melting conditions).

**R1C9:** We mean that the pixels with SIC above 80% for less than 5 days (i.e., very short-lived sea ice) were not included in the analyses. That is to say, these pixels were marked as being ice-free (Markus et al., 2009).

- Sea ice pixels and the occurrences of sea ice were first determined, and the melt detection methods were applied henceforth. We employed the same preconditions for melt detection on sea ice based on AMSR-E/2 and ERA. This may not result in the difference in MDF retrieved from the two methods.
- This sentence was rephrased to clarify:

  Pixels with SIC greater than 80% for less than 5 days were marked as being ice-free (Markus et al., 2009). For a sea ice pixel, SIC above 15% indicates the presence of sea ice (Meier and Stroeve, 2008).

P4, L20: please state the exact field of the ERA interim dataset used including timestep, are you using "Air temperature at 2m height" from the surface analysis? Also: in how far was the data used to "assist" with the AMSR-E/2 melt detection? Is this is described somewhere else in the text?

**R1C10:** We revised this sentence to clarify:

The 6-hourly air temperature (Tair) from the gridded ERA-Interim reanalysis at 2 m height was used to...

Yes, in Section 3.2 (Melt detection methods), we clearly described how the ERA-Interim Tair was used.

- First, ERA-Interim Tair was used to assist with melt detection based on AMSR-E/2:

    Further, melt detection was constrained to the days with compatible thermal regimes following Belchansky et al. (2004). The days with ERA-Interim Tair > -5°C were first determined, and the DAV36 algorithm was applied henceforth.

- Second, ERA-Interim Tair was also used to determine snowmelt directly:

    To evaluate the performance of the DAV method on a larger scale, snowmelt over the pan-Antarctic was also determined by ERA-interim reanalysis when the daily maximum Tair exceeded -1°C.

P4, L29-P5: It is unclear what the MEMLS simulation is for. In Kang et al. (2014), which you are citing four times in this paragraph, this is discussed in very detail. The variation of snow grain size is barely discussed in this paragraph and from what I got, never really picked up again in the manuscript. I would probably just remove the Fig. 1.

**R1C11:** We recalled the work from Kang et al. (2014) to show that the DAV method is superior to single-channel methods in snowmelt detection. In Willmes et al. (2009), the DAV method was only applied in the detection of snowmelt onset on sea ice. The analysis with varying snow grain size was added to explain that the DAV method can be used to detect snowmelt throughout the melt season with snow metamorphism:

- In early melt season, $Tb_{V36}$ of the fine-grained snowpack increases rapidly in energy saturation phase with a slight amount of liquid water. Daily Tb variations are large because of the contrasting freeze/thaw state. DAV method can recognize these sharp changes.

- During the melt seasons, snow grain size can increase to 2 mm when meltwater refreezes in the pore space (Winebrenner et al., 1994). $Tb_{V36}$ from a melting snowpack may be even lower than the winter mean due to the enhanced volume scattering, and single-channel methods may fail to work (Zheng et al., 2018). By contrast, significant daily $Tb_{V36}$ variations still exist in the transition from dry to wet snow regime in the coarse-grained snowpack, and the DAV method still works.

Fig. 1 illustrates the advantage and principle of DAV method in melt detection. We revised this section to clarify the necessity of this figure:

    $Tb_{V36}$ of the fine-grained snowpack increases rapidly in the energy saturation phase with a slight amount of liquid water, and daily Tb variations are large because of the contrasting freeze/thaw

state. Therefore, both single-channel and DAV methods can recognize these sharp changes in the early melt season. During the melt seasons, snow grain size can increase to 2 mm when meltwater refreezes in the pore space (Winebrenner et al., 1994). As a result, $Tb_{V36}$ of the coarse-grained snowpack is much lower than that of the fine-grained snowpack due to enhanced volume scattering. Single-channel methods may fail to work when the $Tb_{V36}$ of a melting snowpack is even lower than the winter mean in the late melt season (Zheng et al., 2018). By contrast, significant daily $Tb_{V36}$ variations still exist in the transition from a dry to wet snow regime during the heavy melt season, even when day-time $Tb_{V36}$ is in the energy dampening phase (Fig. 1). Diurnal freeze-thaw cycles are prevalent in polar regions (Hall et al., 2009; Willmes et al., 2009; van den Broeke et al., 2010b). The simulations suggest that the DAV method can detect melt signals for both the melt onset (e.g., Willmes et al., 2009) and the entire melt season when diurnal freeze/thaw transition occurs (e.g., Zheng et al., 2018). Moreover, the optimum acquisition time of AMSR-E/2 enables us to take full advantage of the DAV method in melt detection.

P5, L5-6: specify the interface you are talking about, probably the snow-air-interface

**R1C12:** Done! Yes, we mean the snow-air interface.

P5, L10-15: Since you employ the method, can you show that this signal is consistent characteristic for melt? For a longer constant melt under full sun illumination, there is probably not much difference between day and night wetness in the snow. Also in Fig. 3, under constant positive air temperatures, there is not constant DAV>10 which indicates that the melt indicator from DAV and the positive temperatures are not strictly connected.

**R1C13:** Yes, positive Tair can only provide evaluation rather than validation of the melt detection methods. Considering the absence of in-situ snow wetness measurements in polar regions, positive Tair was always used to evaluate satellite-derived snowmelt because the occurrence of surface melt corresponds to the spatial pattern of Tair (Tedesco, 2009; Liang et al., 2013). However, melt signals (i.e., the presence of snow liquid water) detected by AMSR-E/2 do not always strictly connect with positive Tair for the following reasons:

- First, positive Tair was derived from the hourly Tair measurements from AWS, while AMSR-E/2 only provide twice-daily observations, which may miss the time when melt occurs.

- Second, passive microwave sensors detect liquid water rather than snowmelt. It takes time for meltwater to refreeze after intense melt events. Subsurface liquid water may remain after the refreezing of the surface, and can still be detected by the satellites due to the penetrating ability of microwave (Zheng et al., 2019).

- Third, owing to the penetration and absorption of solar radiation within the snowpack, snowmelt may occur when Tair is below the freezing point (Koh and Jordan, 1995).
- Last, the DAV method may fail to detect snowmelt when liquid water does not refreeze or snowpack is still melting in warm nights (Willmes et al., 2009).

5 In the revised manuscript, we explained the reasons why satellite-derived melt signals and the positive Tair are not strictly connected:

Melt signals derived from the DAV method and the positive Tair from AWS were not always strictly connected (Figs. 3&4). Their differences may be attributed to inconsistent temporal resolutions because snowmelt and refreezing can occur at any time of the day. The daily
10 maximum Tair was derived from hourly Tair records, while only two daily satellite observations were used in the DAV method. In addition, snowmelt may occur when Tair is below the freezing point because of the penetration and absorption of solar radiation within the snowpack (Koh and Jordan, 1995).

The limitations of the DAV method were also clarified:

15 There are several uncertainties in the pan-Antarctic snowmelt derived from AMSR-E/2 data. First, the DAV method may fail to work when liquid water does not refreeze or snowpack is still melting in warm nights (Willmes et al., 2009). The regions with snowmelt that became more prevalent would presumably show a decrease in melting days based on the DAV method. Fortunately, unlike the Arctic, surface snowmelt on the Antarctic sea ice is always patchy and
20 relatively short-lived (Drinkwater and Liu, 2000). Second, although the DAV method used in this study performs well when compared with meteorological observations, the optimal threshold may differ temporally and regionally with varying snow properties. In addition, ice disintegrates, brine and flooding effects may play an important role in seasonal and even diurnal sea ice Tb variations, further complicating the story (Smith, 1998; Willmes et al., 2009).

**R1C14:** The differences between AMSR-E $Tb_{V36}$ and SSM/I $Tb_{V37}$ are small (Dai and Che, 2010). The comparisons between SSM/I and AMSR-E Tb observations in the dry snow zone suggest the effect of other influences is very limited (Fig. R2). AMSR-E/2 DAV36 is superior to SSM/I DAV37 in melt
30 detection because of the more stable orbit and more appropriate acquisition time. Please see **R1C3** and **R1C8** for full details.

**R1C15:** The differences between AMSR-E Tb$_{V36}$ and SSM/I Tb$_{V37}$ are small (Dai and Che, 2010). The comparisons between SSM/I and AMSR-E Tb observations in the dry snow zone suggest the effect of other influences is very limited (Fig. R2). AMSR-E/2 DAV36 is superior to SSM/I DAV37 in melt detection because of the more stable orbit and more appropriate acquisition time. Please see **R1C3** and **R1C8** for full details.

P5. L31: I cannot see this in Figure 2. There are at least 3 years (2005, 2007,2009) where there is day in mid-winter with positive air temperature where DAV does not exceed the threshold nor shows any signal.

**R1C16:** Positive Tair and snowmelt are short-lived in winter. Positive Tair was derived from the hourly Tair measurements from AWS, while AMSR-E/2 only provide twice-daily observations, which may miss the short-lived melt events because refreezing can be quasi-instantaneous in the Antarctic (van den Broeke et al., 2010a).

Positive Tair can only provide evaluation rather than validation of the melt detection methods. Considering the absence of in-situ snow wetness measurements in polar regions, positive Tair was always used to evaluate satellite-derived snowmelt because the occurrence of surface melt corresponds to the spatial pattern of Tair (Tedesco, 2009; Liang et al., 2013). However, melt signals (i.e., the presence of snow liquid water) detected by AMSR-E/2 do not always strictly connect with positive Tair. We have explained the reasons for their differences in **R1C13**.

P5. L32: Accuracy and Kappa should be defined somewhere.

**R1C17:** The overall accuracy and Kappa coefficient were defined in the revised manuscript:

The overall accuracy ($p_0$, the proportion of observed agreement) and Kappa coefficient $k = (p_0 - p_c)/( 1- p_c)$ were used to measure the coincidence based on the confusion matrix, where $p_c$ is the proportion in agreement due to chance (Cohen, 1960).

P6. L9 (Eq 4): This is only true under the assumption that SIC did not change within the_12h from ascending to descending overflight. This should be mentioned. The method could be optimized in this regard by using the Tbs to retrieve the ice concentration in ascending and descending separately and then calculate the DAV with the aid of an open water tie point (which does not cancel out in case SICs are different for ascending and descending overflights)

**R1C18:** We thank the referee for pointing out this issue.

Yes, the equation is true under the assumption that SIC is the same for both passes. We revised this part to clarify:

Regardless of the atmospheric effects, the Tb of sea ice is comprised of the ice portion ($Tb_{ice}$) and open water portion ($Tb_{ow}$) (Markus and Cavalieri, 1998):

$$Tb = Tb_{ice}\ SIC + Tb_{ow}\ (1-SIC) \tag{3}$$

therefore, $DAV36_{ice}$ can be calculated as follows:

$$DAV36_{ice} = \left| \frac{Tb_{V36A} - Tb_{ow}(1-SIC_A)}{SIC_A} - \frac{Tb_{V36D} - Tb_{ow}(1-SIC_D)}{SIC_D} \right| \tag{4}$$

where $SIC_A$ and $SIC_D$ represent the SIC for ascending and descending passes. If we assume that the SIC of the two passes remains unchanged (i.e., $SIC_A = SIC_D$), then we have:

$$DAV36_{ice} = \frac{|Tb_{V36A} - Tb_{V36D}|}{SIC} \tag{5}$$

We acknowledge the $DAV36_{ice}$ algorithm can be further improved with corresponding SIC for each pass. However, producing a twice-daily SIC product is challenging at present. Extensive simultaneous ground- and space-based observations are needed in the validation. This is likely to be difficult to achieve in polar regions and beyond the scope of this paper. We mentioned this issue in Section 5.2 (Uncertainties):

The DAV36ice algorithm for sea ice snowmelt detection assumes that the SIC of the two passes remains unchanged, which may not be true and lead to misidentifications of melt signals due to quick sea ice drift and disintegration. The algorithm can be further improved if the twice-daily SIC product is available in the future.

P6. L20: accuracy and Kappa definition again

**R1C19:** Done!

P6. L22: Why is a spatial median needed here, what are erroneous microwave signals?

**R1C20:** We explained why a spatial median is needed:

Spurious Tb variations may occasionally be mistaken for melt signals, which can be caused by clouds, atmospheric water vapor, wind-induced surface roughening, and residual calibration errors. To mitigate their impacts on melt detection, a median filter with a 3×3 window was applied to the satellite observations.

P6. L25: "Melt freeze-up and duration…." - I don't understand what is meant here

**R1C21:** Some studies include the analyses of freeze-up (the last day with surface snowmelt) and duration (the days between melt onset and freeze-up) on sea ice (e.g., Markus et al., 2009). However, sufficient Antarctic sea ice melts quickly in austral summer and does not emerge any more in the melting year.

In such cases, the last day with snowmelt is always the day that sea ice disappears, rather than the day that freeze-up begins. We revised this sentence to clarify:

> Sufficient Antarctic sea ice melts quickly in austral summer and does not emerge again in the melting year. In such cases, the last day of snowmelt is always the day that sea ice disappears, rather than the day that freeze-up begins. Thus freeze-up and melt duration were not included in this study.

P6. L29: extend -> extent

**R1C22:** Done!

P7. L7: if below -5_C means frozen state (P6. L24) and above -1_C means melting, what state is there in between and how is that classified?

**R1C23:** There is no intermediate state in freeze/thaw cycles. The two conditions were used separately and do not contradict each other:

- The first condition was used to mitigate the effect of spurious Tb variations (see **R1C20**) in melt detection based on AMSR-E/2.
- The second condition was used to derive snowmelt directly based on ERA-Interim Tair.

The first condition was not used to determine the freeze/thaw state. To avoid confusion, we rephrased this sentence:

> Further, melt detection was constrained to the days with compatible thermal regimes following Belchansky et al. (2004). The days with ERA-Interim Tair > -5°C were first determined, and the DAV36 algorithm was applied henceforth.

P7. L16: Discussion about Fig. 5, see also General 2.: the mid July melt onset around -60 to -65 latitude is quite surprising and needs discussion. Also the later melt onset in the more outer parts are interesting. Is it because there was no ice at the melt onset of the more southern regions and ice drifted there later so that melt occurs later in these regions? However, than the MDF should be even higher in these regions, probably close to 100%. I would also suggest not using the parula but a diverging colormap for the difference plot.

**R1C24:** We thank the referee for pointing out this issue. Yes, this is because sea ice advance (the first day when SIC > 15%) in these areas is later than July 1 (the first day of melting year).  Specifically, sea ice does not occur until early September in some parts of the marginal sea ice zone (Stammerjohn et al., 2008) (Fig. R4).

[Figure]

Figure R4. Day of Antarctic sea ice advance over 1979–2004. The figure is adopted from Stammerjohn et al. (2008).

Therefore, early melt onset (EMO) on some marginal sea ice is later than that on the sea ice in lower altitudes where transient melt events can occur before September (e.g., Fig. R1c).

We mentioned this issue in the revised manuscript:

> In some parts of the marginal sea ice zone, EMO was later than that in higher altitudes. This is because sea ice did not occur in these regions until early September (Stammerjohn et al., 2008), while transient surface snowmelt can occur before that in August at lower latitudes (Supplement Fig. 2). However, the earliest CMO was still found in the marginal sea ice zone (Fig. 5b,d).

Fig. 5 was also redrawn as suggested.

P7. L31: "Fig. 5k-o" -> "Fig. 5g-i"

**R1C25:** Done.

P8. Discussion about Fig. 6: I would suggest splitting the histograms to maybe a 7 by 3 plot to be able to discuss the particular regions better. Also bin size in the histograms is too small, i.e., the histograms are to noisy to comfortable read their data.

**R1C26:** Fig. 6 was redrawn as suggested.

[Figure]

P8. L11-12: The comparison of the melt extent of AIS with the sea ice area makes no sense in my opinion. The AIS is a much smaller region. What is the purpose of this comparison? I suggest to remove Figure 7 completely. However, the melt extentis quite small in the early months like July/August on sea ice which actually contradicts the early melt onset in Fig. 6. This also indicates that the early melt onset is probably just a random noise effect since it does not cover a large area apparently.

**R1C27:** We thank the referee for pointing out this issue.

- We agree that the comparison between melt extent and sea ice extent makes little contribution to this work. The comparison and Fig.7a were removed, but we would like to keep Fig. 7b which illustrates the seasonal evolution of surface snowmelt. In addition, melt extent fraction (MEF) will

be further discussed in Section 5.3 (Response of the pan-Antarctic surface snowmelt to atmospheric indices).

- The occurrence of early snowmelt onset (EMO) does not result in a significant increase in melt extent because the early melt events are always short-lived. Instead, the histogram of continuous melt onset (CMO) suggests most of the pan-Antarctic continuous snowmelt began between October and January (Fig. 6) when MEF increased quickly (Fig. 7).

- Snowmelt can occur in austral winter on both Antarctic ice sheet (Kuipers Munneke et al., 2018) and the Antarctic sea ice (Massom et al., 2001). In Fig. R1c, winter melt events have been clearly observed based on both AMSR-E and SSM/I measurements, accompanied by positive Tair. Yes, these early melt events are always short-lived, and can be easily regarded as random noises. We have conducted data preprocessing and quality control to mitigate the effect of spurious Tb variations on melt detection:

  Spurious Tb variations may occasionally be mistaken for melt signals, which can be caused by clouds, atmospheric water vapor, wind-induced surface roughening, and residual calibration errors. To mitigate their impacts on melt detection, a median filter with a 3×3 window was applied to the satellite observations. Further, melt detection was constrained to the days with compatible thermal regimes following Belchansky et al. (2004). The days with ERA-Interim Tair > -5°C were first determined, and the DAV36 algorithm was applied henceforth.

P8. L17: with "mean maximum MEF" you mean the "Mean anual Maximum MEF" right? should than be changed in the text.

**R1C28:** We mean the maximum of daily mean MEF, this part was removed as suggested (see **R1C27**).

P9. L2: I actually do not understand how the trends are calculated. Fig 9 indicates that you calculate the trends pixel based. One would expect that neighbouring pixel having similar melt onset dates (Fig 9a). If the shown pixel based trends have any significance also the spatial pattern should be coherent.

**R1C29:** Yes, the trends were calculated for each pixel.

- There are a large number of transient melt events during the melt seasons (Fig. 6). Melt onset always shows considerable spatial and temporal variations. A similar phenomenon can be found in the Arctic (Fig. R5) (Stroeve et al., 2014).

[Figure]

Figure. R5. Trends in Arctic sea ice snowmelt onset from 1979 to 2013. This figure is adopted from Stroeve et al (2014).

- Actually, most of the trends were not statically significant. We redrew Fig.9 with additional significance levels of the trends.

P11. L 18: I suspect the values and discussion to change in case you reconsidered the early snowmelt onset

**R1C30:** We assume the referee would like to see the analyses and discussion of continuous melt onset (CMO), which were included in the revised manuscript.

**Author Response to Referee #2**

This study makes use of passive microwave data from AMSRE and AMSR2 to detect melt over the Antarctic Ice Sheet and sea ice regions using a diurnal difference in brightness temperature algorithm. Means and trends in melt onset, number of melt days, and melt day fractions from 2002-2017 are presented and compared with ERA estimates of surface melt based on air temperature, and SSMI melt indices. A method of improving melt detection in marginal sea ice is also presented and validated.

In general the paper is well written and of great interest with excellent figures but a few points need to be addressed. There are many instances where the use (or not) of the definite article is incorrect, I suggest a read through by a native English speaker to correct these.

**R2C1:** We are sorry for the grammatical problems and the inconvenience they caused in reading. The manuscript was thoroughly revised and edited by a native speaker. We hope it can meet the journal's standards.

Early on it should be made clear that satellite algorithms for melt retrieval detect either the presence or absence of liquid water, or the diurnal transition between the two, rather than actual melting events.

**R2C2:** Yes, microwave sensors only detect snow liquid water rather than snowmelt. We clarified this issue in the revised manuscript:

> Therefore, snowmelt can be detected via microwave radiometry by identifying the sharp changes in microwave brightness temperatures (Tb) caused by the presence of snow liquid water (Serreze et al., 1993; Liu et al., 2005).

Various products used in this study, Tb, Tair, SIC were undoubtedly supplied at different projections and swaths and resolutions. Please provide more detail on how these products were coregistered.

**R2C3:** We explained how these products were coregistered:

> The sea ice product is provided in the NSIDC EASE-Grid projection, which is the same as the AMSR-E/2 products. The 0.5° gridded ERA-Interim reanalysis was reprojected to the NSIDC EASE-Grid, and resampled to the same spatial resolution as the passive microwave measurements (25 km).

The validation described briefly P5, L28 does not give enough detail. What is the 'melt signal determined by satellite'? How are the accuracy and coefficients referred to calculated? Why is this agreement 'in contrast'? In contrast to what?

**R2C4:** Melt signal determined by satellite is actually the presence of snow liquid water. The overall accuracy and Kappa coefficient were clearly defined. "By contrast" was changed to "However". The evaluation method was described in more details to clarify these issues:

> The verification of snowmelt is difficult, especially in the pan-Antarctic where meltwater refreezes quickly, and climatic data are sparse. However, surface snowmelt is determined by atmospheric conditions and agrees well with the Tair distribution pattern (Tedesco, 2007; Liang et al., 2013). In-situ Tair measurements at Zhongshan Station (69.37°S, 76.38°E) obtained from the Chinese National Arctic and Antarctic Data Center (www.chinare.org.cn) were used to evaluate the DAV36 algorithm (Fig. 2). The overall accuracy ($p_0$, the proportion of observed agreement) and Kappa coefficient $k = (p_0 - p_c)/(1 - p_c)$ were used to measure the coincidence based on the confusion matrix, where $p_c$ is the proportion in agreement due to chance (Cohen, 1960). $Tb_{V36A}$ and $Tb_{V36D}$ showed sharp increases at melt onset, while decreased below the winter mean in the late melt seasons with associated snow metamorphism. However, positive maximum Tair agreed well with melt signals (i.e., the presence of liquid water) determined by the DAV method, with an overall accuracy of 0.93 and a Kappa coefficient of 0.79.

P1, L21. It does not make sense to compare snow melt extents of sea ice and ice sheets when they cover different areas in total. What is the point?

**R2C5:** We agree that the comparison between melt extent and sea ice extent makes little contribution to this work. The comparison and Fig.7a were removed, but we would like to keep Fig. 7b which illustrates the seasonal evolution of surface snowmelt. In addition, melt extent fraction (MEF) will be further discussed in Section 5.3 (Response of the pan-Antarctic surface snowmelt to atmospheric indices).

P1, L28. You mean snow melt leads to an increase in size of snow grains.

**R2C6:** Good catch. We rephrased this sentence:

> Intense snowmelt leads to the formation of melt ponds on sea ice and ice sheets, which in turn absorb more radiation and induce further snowmelt through melt-albedo feedback (Tanaka et al., 2016; Bell et al., 2018).

P1, L30. You confuse ice sheets on bedrock, with the hydrofracture on ice shelves which are floating. Separate the discussion of these two impacts.

**R2C7:** Revised as suggested:

> Meltwater may fill in the ice crevasses on ice sheets and migrate to the ice-bedrock surface, which can induce the acceleration of ice flow (Zwally et al., 2002; Sundal et al., 2011). Meltwater can also transport heat into crevasses and deepen them, providing the conditions for ice shelves to break up through hydrofracturing (Scambos et al., 2000; van den Broeke, 2005).

P3, L13. It needs to be made clearer why DAV is more likely to detect melt with AMSRE/2. Ie Explain why time of day (rather than period) of the overpasses is important.

**R2C8:** The acquisition time is important for melt detection for two reasons:

- First, measurements with stable acquisition time are superior in the analyses of inter-annual snowmelt dynamics. AMSR-E and AMSR2 operate in controlled-orbits measurements, and the crossing time for the two sensors are nearly the same. By contrast, crossing time differs between SSM/I sensors and also changes significantly over the years of operation due to orbit degradation (Picard and Fily, 2006).
- Second, the Antarctic diurnal melt area varies approximately as a sinusoid with the peak in the afternoon and the trough in the early morning (Picard and Fily, 2006). Compared with SSM/I, AMSR-E/2 have more opportunities to detect melt events in the pan-Antarctic due to warmer and colder periods for ascending and descending passes and an expected higher DAV. Taking 2002-2003 as an example, the local acquisition time of ascending and descending SSM/I Tb product south of 40° S were 19.17±0.44 and 5.45±0.45, while they were 14.16±0.20 and 0.88±0.20 for AMSR-E Tb product.

We revised this paragraph to make it clear:

> Most of these studies investigated surface snowmelt on sea ice and ice sheets based on SSM/I sensors. However, SSM/I observations show considerable variations in local acquisition time because of orbit degradation (Picard and Fily, 2006). By contrast, the Advanced Microwave Scanning Radiometer for the Earth Observing System (AMSR-E) and the Advanced Microwave Scanning Radiometer 2 (AMSR2) operate in controlled-orbits so that local acquisition time shows little temporal variation (http://www.remss.com/support/crossing-times). AMSR-E/2 measurements with a stable orbit are superior in the analyses of inter-annual snowmelt dynamics. Diurnal melt area in the Antarctic varies approximately as a sinusoid with the peak in the afternoon

and the trough in the early morning (Picard and Fily, 2006). AMSR-E/2 can monitor the Antarctic sea ice and ice sheet (referred to as pan-Antarctic) surface snowmelt at the appropriate local acquisition time. Taking 2002-2003 as an example, the local acquisition time of ascending and descending SSM/I Tb products south of 40° S were 19.17±0.44 and 5.45±0.45, respectively, while these values were 14.16±0.20 and 0.88±0.20 for the AMSR-E Tb products. Compared with SSM/I, AMSR-E/2 have more opportunities to detect melt events in the pan-Antarctic due to warmer and colder periods for ascending and descending passes and an expected higher DAV.

P3, L21. 'Meltwater on the AIS always refreezes instantaneously'. Needs a reference. Also,in this case, it would never be detected.

**R2C9:** We thank the referee for pointing out the mistake. We mean meltwater can refreezes quasi-instantaneously. We revised this sentence with appropriate citations:

Meltwater on the Antarctic ice sheet (AIS) can refreeze quasi-instantaneously (van den Broeke et al., 2010a) and contributes little to the surface mass balance (The IMBIE team, 2018).

P4, L7. Changes 'almost shares' to 'shares almost'.

**R2C10:** Done.

P4, L17. This sentence does not make sense. Please rewrite. Which air temperature was used? 2 m?

**R2C11:** Yes, the 2 m Tair from the gridded ERA-Interim reanalysis was used in this study. We rewrote this paragraph:

ERA-Interim is a global reanalysis produced by the European Centre for Medium-Range Weather Forecasts (ECMWF). The ERA-Interim reanalysis includes various surface parameters, describing weather, ocean and land-surface conditions since 1979 (Dee et al., 2011). The 6-hourly air temperature (Tair) from the gridded ERA-Interim reanalysis at 2 m height was used to assist with melt detection based on AMSR-E/2, as well as directly determine snowmelt in this study.

P4, L29. Please move reference to Fig. 1 to later in this paragraph. You have not yet described the simulations.

**R2C12:** Done.

**R2C13:** Done.

5   **R2C14:** Done.

**R2C15:** Done.

**R2C16:** Good catch. We rewrote this sentence:

> In general, snowmelt shows significant latitudinal zonality. EMO and CMO occur later from the marginal sea ice to the inland of the AIS, from low-latitudes to high-latitudes, while MDF
15     increases in the opposite direction.

**R2C17:** This condition was used to mitigate the effect of spurious Tb variations on melt detection based on AMSR-E/2, rather than determine snowmelt.

20   To avoid confusion, we rephrased this sentence:

> Further, melt detection was constrained to the days with compatible thermal regimes following Belchansky et al. (2004). The days with ERA-Interim Tair > -5°C were first determined, and the DAV36 algorithm was applied henceforth.

**R2C18:** Corrected.

**R2C19:** We agree that the comparison between melt extent and sea ice extent makes little contribution to this work. We removed this part in the revised manuscript.

P8, L16. It is not clear why the decreasing sea ice extent would lead to an increasing sea ice melt extent? This would only explain the delayed peak in sea ice MEF.

**R2C20:** We mean the peak of sea ice melt extent does not occur in the warmest period because sea ice considerably declines before January. This part was removed in the revised manuscript.

Fig. 9. You should only plot those pixels with a significant trend. Or also plot the p values.

**R2C21:** Good suggestion. We added black points in Fig.9 to indicate the pixels with significant trends above the 90% confidence level.

P9, L14. You should discuss further implications of the failure of DAV when melt is continuous. This would presumably manifest as a decrease in Melt Days detected where melt temporal continuity became more prevalent. Might this also explain areas with very early melt onset such as in BAS but a surprisingly low number of melt days.

**R2C22:** This is an important point. We added the discussion about this issue in Section 5.2 (Uncertainties):

> The regions with snowmelt that became more prevalent would presumably show a decrease in melting days based on the DAV method. Fortunately, unlike the Arctic, surface snowmelt on the Antarctic sea ice is always patchy and relatively short-lived (Drinkwater and Liu, 2000).

Yes, the earliest melt onset was found in BAS, but the highest melting days (37 days) and MDF (17%) were also found in this region. But the melt onset was very early but not too many melt events were detected in RS. We mentioned this point in the revised manuscript:

> The might also be the reason that the melt onset was very early in RS while few melt events were detected by AMSR-E/2.

P10, L25. Please include these correlations between atmospheric indices and melt in a table.

**R2C23:** We added a table (Table 3) to show the correlations between the atmospheric indices and melt indices.

> Table 3. Correlation between snowmelt index and atmospheric index for the Period 2002–2017.

Correlation coefficients with *, ** and *** indicate statistical significance at 90%, 95%, and 99% confidence levels, respectively.

| Atmospheric index | Melt index | WS | IO | PO | RS | BAS | AIS | All |
|---|---|---|---|---|---|---|---|---|
| SAM | EMO | -0.27 | 0.77*** | 0.32 | 0.16 | 0.25 | 0.52* | 0.31 |
| | CMO | -0.03 | 0.15 | 0.03 | 0.50* | 0.53* | 0.80*** | 0.54** |
| | Melting days | 0.37 | 0.18 | 0.01 | -0.26 | -0.42 | -0.88*** | -0.02 |
| | MDF | 0.18 | -0.11 | -0.07 | -0.48* | -0.53* | -0.88*** | -0.33 |
| SOI | EMO | 0.11 | -0.08 | -0.01 | 0.10 | 0.28 | 0.55** | 0.19 |
| | CMO | 0.09 | -0.23 | 0.14 | -0.53* | -0.12 | 0.18 | -0.31 |
| | Melting days | 0.03 | -0.26 | 0.07 | -0.11 | 0.46* | -0.03 | -0.07 |
| | MDF | -0.16 | -0.18 | 0.12 | 0.15 | 0.34 | -0.03 | -0.09 |
| Nino3.4 | EMO | -0.28 | -0.06 | -0.11 | -0.02 | -0.26 | -0.47* | -0.27 |
| | CMO | -0.15 | 0.19 | -0.35 | 0.66** | 0.28 | 0.01 | 0.35 |
| | Melting days | 0.04 | 0.42 | -0.16 | 0.08 | -0.54** | -0.15 | -0.07 |
| | MDF | 0.24 | 0.36 | -0.11 | -0.22 | -0.46* | -0.15 | 0.13 |

**Author Response to Editor**

5 p6L22 "if we assume": what happens if you don't make this assumption. How much do the results change? What impact does this assumption has on the final outcome.

**EC1:** Open water shows a much lower Tb and may dampen melt signals in marginal sea ice zone. Therefore, we employed $DAV36_{ice}$ (i.e., DAV36 contributed by the ice-covered portion) rather than DAV36 (the difference between AMSR-E/2 ascending and descending 36.5 GHz Tb) in melt detection 10 on sea ice. $DAV36_{ice}$ can be calculated based on the sea ice concentration (SIC) for ascending and descending passes.

- However, no twice-daily SIC product is available now, and the daily SIC product is the best alternative if we assume that the SIC of the two passes remains unchanged.
- $DAV36_{ice}$ can not be solved if we don't make this assumption. We acknowledge this assumption 15 may not be true and lead to misidentifications of melt signals when sea ice drifts or disintegrates quickly, and the $DAV36_{ice}$ algorithm can be further improved with corresponding SIC for each pass. We discussed this issue in the revised manuscript:

The $DAV36_{ice}$ algorithm for sea ice snowmelt detection assumes that the SIC of the two passes remains unchanged, which may not be true and lead to misidentifications of melt signals due to 20 quick sea ice drift and disintegration. The algorithm can be further improved if the twice-daily SIC product is available in the future.

p9 + table 2: please also add uncertainties (+- x days) on the observed trends and indicate how these uncertainties were calculated. If the uncertainties are in the same order of magnitude as the trends, the trends do not mean much.

5 **EC2:** Good suggestion. The uncertainties of the trends were added in the revised manuscript:

Table 2. Trends in pan-Antarctic EMO, CMO, melting days and MDF during 2002-2017. Trends with *, ** and *** indicate statistical significance at 90%, 95%, and 99% confidence levels, respectively. The uncertainties of the trends were estimated at 90% confidence level.

| Melt index | WS | IO | PO | RS | BAS | AIS | All |
|---|---|---|---|---|---|---|---|
| EMO (days yr$^{-1}$) | -0.35±1.10 | 0.09±0.82 | 1.37±1.40 | 1.72±0.71*** | 2.13±1.88* | 0.71±0.76 | 0.68±0.59* |
| CMO (days yr$^{-1}$) | -0.82±0.72* | -0.84±0.51** | -0.16±0.71 | 0.52±1.23 | 0.59±1.74 | 0.22±0.51 | -0.22±0.43 |
| Melting days (days yr$^{-1}$) | 0.34±0.33* | 0.37±0.42 | 0.80±0.46*** | -0.52±0.43* | -0.05±0.57 | -0.33±0.41 | 0.11±0.20 |
| MDF (% yr$^{-1}$) | 0.20±0.14** | 0.15±0.14* | 0.19±0.12** | -0.14±0.14 | 0.02±0.19 | -0.09±0.11 | 0.07±0.08 |

10 Other changes in the revised manuscript include:

- We have made mistakes in statistical analyses in the first version, but they do not affect the main conclusion of this study. We corrected these mistakes and checked the manuscript carefully.
- The manuscript was edited by a native English speaker and the grammatical errors were corrected.
- Figure 5-13 was redrawn, two figures were added as supplements.

[revised manuscript text omitted]

---

## Author Response (AR2)

Dear Dr. Stef Lhermitte,

Thank you so much for the time and attention dedicated to this manuscript and for the valuable comments. Below, we answer point-by-point all the comments. The comments are reproduced in blue and the authors' responses (**AR**) are provided in black. The underlined texts are our corresponding changes in the revised manuscript. Pages (**P**) and lines (**L**) referred to in the response are directed to the revised manuscript with changes highlighted.

The main changes in the revised manuscript include:

- The robustness and the uncertainties of the DAV36$_{ice}$ algorithm were explained.
- The MEMLS simulations (including Figure 1) were removed.
- The area-averaged trend analyses were removed.
- A new section (3.2. Data preprocessing and analysis) was added.
- The comparison between snowmelt and climate indices was moved to the Methodology and Results Section.

Best regards,

Lei Zheng (on behalf of the co-authors)

**Major remarks**

1. I agree with R1 that Figure 1 does not directly conveys a clear message to your paper. I would recommend to either remove it (if Kang already showed that DAV is superior) or redesign (e.g. by showing a time series of single channel vs. DAV for different types of snowpacks) it such that it is clear that DAV method is superior. Currently, it is not very clear what the added value within the story is.

**AR1**: Figure 1 and the MEMLS simulations were removed. Instead, we cited the work which shows that DAV method is superior to the single-channel method:

> Single-channel methods may fail to work when the Tb of a melting snowpack is even lower than the winter mean in the late melt season. By contrast, significant DAV variations still exist in the transition from a dry to wet snow regime during the heavy melt season. In addition, DAV is very sensitive to snowmelt and show very little variation in frozen seasons (Zheng et al., 2018). **(P5L20-24)**

2. Accuracy of the method: you indicate accuracies/Kappa-coefficients between Tair and satellite melt of 0.79 and 0.51 (for DAV) and 0.82 and 0.60 (for DAV_ice). Based on these values I doubt if these are strong enough to warrant any later interpretation of the trends (especially for the more chaotic EMO (see next point))? Please clarify this and take this uncertainty along if you later want to use the DAV signal for melt interpretations

**AR2**: The DAV36$_{ice}$ algorithm was utilized to detect melt signals on sea ice by mitigating the effect of open water. Actually, melt detection methods in polar regions have never been validated because of the absence of continuous and extensive snow wetness measurements. Tair can only provide evaluation rather than validation of the melt detection methods. The overall accuracy and Kappa coefficient between the positive Tair and satellite-detected melt signals were 0.82 and 0.60 (not 0.79 and 0.51), respectively.

The satellite-detected (25 km) and station-derived thaw/freeze state are not always strictly connected. For example, the overall accuracy between soil thaw/freeze status determined from satellite and ground temperature lies in the range of 0.79 to 0.92 (Jin et al., 2009; Zhao et al., 2011; Derksen et al., 2017; Kim et al., 2017). An overall accuracy of 0.82 between the thaw/freeze status estimated from microwave satellite and buoy Tair is acceptable. Moreover, a Kappa coefficient of 0.60 represents a fair to good strength of agreement (Fleiss et al., 2003). This is confirmed by the obtained p-value ($p < 0.05$). The mismatches largely occur in the period with transient snowmelt when thaw/freeze status shifted quickly and in the regions with plenty of open water (typical SIC < 90%). For the periods that were identified as thaw/freeze status for at least three days, satellite-derived melt signals showed close agreement (OA=0.87, Kappa=0.68) with that derived from Tair over the sea ice with SIC > 90%.

We explained the robustness of the DAV36$_{ice}$ algorithm in the revised manuscript:

> …while those were 0.82 and 0.60 based on the DAV36$_{ice}$ algorithm, indicating a fair to good strength of agreement according to Fleiss et al. (2003). (**P7L14-16**)
> The mismatches largely occur in the period with transient snowmelt when thaw/freeze status shifted quickly and in the regions with plenty of open water (typical SIC < 90%). For the periods that were identified as thaw/freeze status for at least three days, satellite-derived melt signals showed close agreement (OA=0.87, Kappa=0.68) with that derived from Tair over the sea ice with SIC > 90%. (**P7L19-22**)

We also explained and the reasons for the disagreements:

- First, the inconsistent temporal and spatial resolutions may result in disagreements in the period with transient melt events.

The disagreements in the identification of transient melt events may result from inconsistent spatial and temporal resolutions. Validating the large-scale satellite-based estimation (25 km for AMSR-E/2) against station-based measurements is always challenging due to representativeness errors (Lyu et al., 2017). The daily maximum Tair was derived from hourly Tair records, while only two daily satellite observations were used in the DAV method. (**P7L22-25**)

- Second, though the DAV36$_{ice}$ algorithm can mitigate the effect of open water, we acknowledge that melt detection over the sea ice is still challenging because of the ice disintegrates, brine and flooding. We have clarified this issue:

  In the marginal sea ice zone, ice disintegrates, brine and flooding effects may play an important role in seasonal and even diurnal sea ice Tb variations, further complicating the story (Smith, 1998; Willmes et al., 2009). (**P7L25-27**)

In view of this, we removed the trend analysis of EMO in the revised manuscript:

EMO is not included in the trend analysis because of the chaotic spatial distribution of the early transient melt events. (**P10L31-32**)

3. I agree with R1 that the early melt onset seems very much random early effect with little physical meaning and/or impact. Therefore, I wonder if the regression on these time series have any physical meaning? I would definitely consider removing the trend analysis of the EMO

**AR3**: We thank the editor for the suggestion. The trend analysis of the EMO was removed.

4. You only introduce the trend regression in the results (which should already be done in the methods) and it is completely unclear how the trends (already indicated by R1) were observed and where their uncertainty comes from. Is it on a per-pixel basis and afterwards spatially merged (and how?) or first spatial merging and then trend-analyses? How were the uncertainties calculated, etc?

**AR4**: Trends at every pixel with continuous records were firstly calculated, and the area-averaged trends were afterwards spatially merged. The trends were reported in days per year, with uncertainties determined using the degrees of freedom present in the regression residuals at the 90% confidence level.

We introduced how the trends were calculated in the Methodology Section:

Linear trends were calculated using a least-squares adjustment (**P9L7-8**)

The area-averaged trend analyses were removed according to the editor's suggestion in the next comment (please see **AR5** for full details).

5. I have my severe doubts about the physical meaning of the trends (especially for the EMO, which seems very stochastic (as also indicated by R1)). I would reconsider removing all trend analyses from the paper and or add more clarity on the methods (how did you do the patio-temporal averaging?)

**AR5:** The area-averaged trend analyses (including Table 2 and Fig. 8) were removed in the revised manuscript, but we would like to keep the trend analyses at pixel-scale (excluding EMO) which can show the localized changes in snowmelt timing. We explained how the trend and significance level were calculated in Section 3.2 (Data preprocessing and analysis):

> Linear trends were calculated using a least-squares adjustment. The significance levels of the correlations and regressions were determined using the Student's t-tests. (**P9L7-9**)

6. The comparison with the atmospheric indices should already be introduced in the results/methods sections and not left until the discussion.

**AR6:** We thank the editor for the suggestion. The comparison was moved to the Methodology Section and Results Section. Additionally, we explained how climate indices were calculated:

> To study the response of the pan-Antarctic surface snowmelt to circulation conditions, we explored the relationship between the melt indices and El Niño/Southern Oscillation (ENSO) and Southern Annular Mode (SAM). Nino3.4, the average sea surface temperature anomaly in the tropical (170°W–120°W, 5°S–5°N), is calculated from the Hadley Centre Sea Ice and Sea Surface Temperature dataset (Rayner et al., 2003). Southern Oscillation Index (SOI) is calculated using the pressure difference between Tahiti and Darwin (Ropelewski et al., 1987). Southern Annular Mode is calculated using the zonal pressure difference between 40°S and 65°S  (Marshall, 2003). (**P9L10-15**)

**PDF Comments**

7. P1L21-22. Given the large intermittent character of EMO, I would remove your trend analyses from the paper.

**AR7:** Trend analysis of EMO was removed as suggested.

8. P3L11-12. Based on the sentence I derive that the units of these values are time, but I really have difficulty the numbers here as time. Please use hour+minutes (or seconds) as unit and modify the values accordingly.

**AR8:** The values were changed to hour±minutes:

> …the local acquisition time of ascending and descending SSM/I Tb products south of 40° S were 19:10±26 minutes and 5:27±27 minutes, respectively, while these values were 14:09±12 minutes and 0:53±12 minutes for the AMSR-E Tb products. (**P3L11-13**)

9. P4L14-15. Delete "University of Colorado Boulder".

**AR9:** Done.

10. P4L15. I guess that you mean over the data whole period and that this is a way to remove all ice free zones. If so, please indicate that it is over the whole period.

**AR10:** Yes. We clarified this issue:

> To remove ice-free zones over the whole period, pixels with SIC greater than 80% for less than 5 days were not included in the analyses (Markus et al., 2009). (**P4L17-18**)

11. P4L22. What resampling scheme was used? Nearest neightbor, linear, .....

**AR11:** Nearest-neighbor interpolation was used. We clarified this issue:

> … and resampled to the same spatial resolution as the passive microwave measurements (25 km) using nearest-neighbor interpolation. (**P4L25-26**)

12. P7L3. "0.79 and 0.51". Is that good enough? Please take this uncertainty along in your later stages.

**AR12:** The DAV36$_{ice}$ algorithm was utilized in melt detection over sea ice rather than the DAV algorithm. The overall accuracy and Kappa coefficient between the positive maximum Tair and satellite-detected melt signals based on the DAV36$_{ice}$ algorithm used in this study were 0.82 and 0.60 respectively.

The satellite-detected (25 km) and station-derived thaw/freeze state are not always strictly connected. For example, the overall accuracy between soil thaw/freeze status determined from satellite and ground temperature lies in the range of 0.79 to 0.92 (Jin et al., 2009; Zhao et al., 2011; Derksen et al., 2017; Kim et al., 2017). An overall accuracy of 0.82 between the thaw/freeze status estimated from microwave satellite and buoy Tair is acceptable. Moreover, a Kappa coefficient of 0.60 represents a fair to good strength of agreement (Fleiss et al., 2003). This is confirmed by the obtained p-value ($p < 0.05$). The mismatches largely occur in the period with transient snowmelt when thaw/freeze status shifted quickly and in the regions with plenty of open water (typical SIC < 90%). For the periods that were identified as thaw/freeze status for at least three days, satellite-derived melt signals showed close agreement (OA=0.87, Kappa=0.68) with that derived from Tair over the sea ice with SIC > 90%.

We explained the robustness of the DAV36$_{ice}$ algorithm in the revised manuscript:

> …while those were 0.82 and 0.60 based on the DAV36$_{ice}$ algorithm, indicating a fair to good strength of agreement according to Fleiss et al. (2003). (**P7L14-16**)
>
> The mismatches largely occur in the period with transient snowmelt when thaw/freeze status shifted quickly and in the regions with plenty of open water (typical SIC < 90%). For the periods that were identified as thaw/freeze status for at least three days, satellite-derived melt signals showed close agreement (OA=0.87, Kappa=0.68) with that derived from Tair over the sea ice with SIC > 90%. (**P7L19-22**)

We also explained and the reasons for the disagreements:

- First, the inconsistent temporal and spatial resolutions may result in disagreements in the period with transient melt events.

  The disagreements in the identification of transient melt events may result from inconsistent spatial and temporal resolutions. Validating the large-scale satellite-based estimation (25 km for AMSR-E/2) against station-based measurements is always challenging due to representativeness errors (Lyu et al., 2017). The daily maximum Tair was derived from hourly Tair records, while only two daily satellite observations were used in the DAV method. (**P7L22-25**)

- Second, though the DAV36$_{ice}$ algorithm can mitigate the effect of open water, we acknowledge that melt detection over the sea ice is still challenging because of the ice disintegrates, brine and flooding. We have clarified this issue:

  In the marginal sea ice zone, ice disintegrates, brine and flooding effects may play an important role in seasonal and even diurnal sea ice Tb variations, further complicating the story (Smith, 1998; Willmes et al., 2009). (**P7L25-27**)

In view of this, we removed the trend analysis of EMO in the revised manuscript:

   EMO is not included in the trend analysis because of the chaotic spatial distribution of the early transient melt events. (**P10L31-32**)

**13. P7L6.** "complete melt seasons". How much disappears as a result of Era-interim?

**AR13:** We are so sorry that we fail to catch what the editor means here. We would appreciate it if the editor could elaborate more on this comment.

[Figure]

**14. P7L9.** "median filter". Spatial? Perhaps add spatial to clearly indicate it is a spatial filter?

**AR14:** We clarified this issue:

   To mitigate their impacts on melt detection, a median **spatial** filter with a 3×3 window was applied to the satellite observations. (**P9L5**)

**15. P7L14.** "was" → "were".

**AR15:** Done.

**16. P7L16.** "disappears". You mean: when it drops below 15% SIC?

**AR16:** Yes, we revised this sentence to clarify:

> In such cases, the last day of snowmelt is always the day that SIC drops below 15%, rather than the day that freeze-up begins. (**P8L12-14**)

17. P8L3. "parts" → "sub-areas"?

**AR17:** Done.

18. P8L6. "AIS" → "including all Antarctic ice shelves".

**AR18:** We revised this sentence to clarify:

> … and the AIS including all floating ice shelves. (**P9L21**)

19. P8L22. Add "...from AMSR-E/2 satellite observations and ERA-reanalyses agreed well...".

**AR19:** Done.

20. P8L23. Is that well? They are often outside each other confidence limits, so I definitely don't agree with 'well'. What is the impact here and how reliable are your results in that 'not that well' context?.

**AR20:** We agree that snowmelt timing derived from the two methods show local discrepancies, especially in the marginal sea ice zone. We revised this sentence:

> Melting timing derived from AMSR-E/2 satellite observations and ERA-Interim reanalyses generally agreed well with each other in the AIS and IO, but showed some discrepancies in other regions (Table 1 and Fig. 4). (**P10L6-7**)

We are sorry to make it confusing here. The same as Markus et al. (2009), we compared the snowmelt from reanalysis Tair and the satellite observations to show that they have similar spatial distribution patterns, rather than to validate the satellite estimations. We clarified this issue in the revised manuscript:

> Melt timing derived from different approaches are supposed to show similar spatial distribution patterns. To evaluate the performance of the DAV method on a larger scale, snowmelt over the pan-Antarctic was also determined by ERA-Interim reanalysis when the daily maximum Tair exceeded -1°C. (**P7L30 & P8L1-2**)

Similar discrepancies between the results from the reanalysis and satellite were also found in the Arctic (Markus et al., 2009):

[Figure]

Figure R1. Examples of melt onset for 2004 from passive microwave (PMW) data and from the NCEP/NCAR and POLES data sets. NCAR and POLES melt data are determined by a threshold in temperature of -1℃. Temperatures above this threshold indicate melt conditions. For the POLES data we distinguish between earliest melt and continuous melt. "Earliest melt" indicates the day when temperatures go above the threshold temperature for the first time while "continuous melt" indicates the day when temperatures remain above the threshold temperature for the summer (figure revised from Markus et al. (2009)).

This is mainly because Tair above/below the freezing point does not always indicate a melting/frozen snowpack. Markus et al. (2009) showed that varying the threshold applied to Tair by ±2℃ can result in a range of ±50 days for melt timing. In addition, it takes time to produce meltwater when melt energy is available. We explained the reasons for the disagreements in the revised manuscript:

Tair above/below the freezing point does not always indicate a melting/frozen snowpack. EMO and CMO derived from ERA-Interim were earlier than that detected by AMSR-E/2 data (Fig. 4 and Table 1). This is because it takes time to produce liquid water when snow temperature approaches the melting point. Daily Tb variation is limited when liquid water does not refreeze or snowpack is still melting in the warm nights during heavy melt seasons (Willmes et al., 2009;

Semmens and Ramage, 2014). As a result, ERA-Interim recognized more melt events for the regions where heavy snowmelt was found and the DAV method fails to work, such as the WS, BAS, RS and the Larsen C ice shelf. This might also be the reason that the melt onset was very early in RS while few melt events were detected by AMSR-E/2. (**P12L17-23**)

21. P9L1. "Approximately16%". Space missing.

**AR21:** Corrected.

22. P9L11. This trend analyses should be introduced already in the method section and it should be definitely be indicated how this was done (how did you spatially combine the pixels, what do you do with the uncertainties, etc). Did you first calculate the regional mean and subsequently applied regression on the time series of means and/or did calculate the trends per pixel and subsequently merged them? Etc..

**AR22:** Trends at every pixel with continuous records were firstly estimated, and the area-averaged trends were afterwards spatially merged. The trends were reported in days per year, with uncertainties determined using the degrees of freedom present in the regression residuals at the 90% confidence level.

We introduced how the trends were calculated in the Methodology Section:

Linear trends were calculated using a least-squares adjustment (**P9L7-8**)

The area-averaged trend analyses were removed according to the editor's suggestion (please see **AR5** for full details).

.

23. P9L20. "Fig. 8". Add titles/labels + uncertainties.

**AR23:** The area-averaged trend analyses (including Fig. 8) were removed as suggested.

24. P9L20. "much later". Is it?.

**AR24:** Yes, EMO over the period 2013-2017 was on average about 12 days later compared with that in 2011/2012.

25. P10L26. Or you may overestimate? I don't see any hard proof that any method is better than the other and you should allow more interpretation of whom is better.

**AR25:** Yes, we cannot tell which method is more accurate given that we did not directly validate the methods and the results from ERA-Interim can only serve as a reference.

We revised this part with more interpretation of the strength and weakness of the two methods:

> AMSR-E/2 and ERA-Interim daily mean melt extent were more than twice the melt extent mapped by PF06 from December to February (Fig. 10b). In a recent study, PF06 melt extent was also found to be much smaller compared with that derived from scatterometers (Zheng et al., 2020). PF06 determined snowmelt when the SSM/I 19 GHz Tb exceeds the winter mean plus 2.5 times the standard deviation of the winter Tb. During the melt seasons, Tb may decrease due to the strong volume scattering resulting from the increase of snow grain size and the formation of icy layers (Ramage and Isacks, 2002; Markus et al., 2009). Summer Tb can even be much lower than the winter observations (Zheng et al., 2018). Therefore, PF06 may underestimate surface snowmelt when snow metamorphism occurs in melt season. The DAV method is unaffected by snow metamorphism but may fail to work when liquid water does not refreeze or snowpack is still melting in warm nights (Willmes et al., 2009). Fortunately, the Antarctic surface snowmelt is always short-lived and can refreeze quasi-instantaneously (Zheng et al., 2018; van den Broeke et al., 2010a). (**P13L5-14**)

26. P11L17. "Response of the pan-Antarctic surface snowmelt to atmospheric indices". Should be a result, not a discussion.

**AR26:** This section was moved to Results.

27. P12L16. "positive Tair observations", of what type of observations?.

**AR27:** Good catch. We revised this sentence to clarify:

> …snowmelt detected by AMSR-E/2 agreed well with the **in-situ** positive Tair observations, and was more extensive than that detected by SSM/I. (**P15L9-10**)

28. P12L18. Avoid abbreviations in the conclusions, so also people that want to have a quick look, can

**AR28:** Abbreviations were changed to full names in the Conclusions.

29. P12L23-26. I agree, but I think this should also be addressed in the discussion and not left for the conclusion.

**AR29:** This part was moved to Section 5.2 (Uncertainties).

**References:**

[revised manuscript text omitted]

---

## Editor Decision (ED2)

[revised manuscript text omitted]

**Figure 2. Meteorological and satellite measurements along a sea ice buoy in the Weddell Sea from 1 September, 2014 to 1 May, 2015. (a) Snow depth (cyan line), daily maximum Tair (pink line), and SIC (olive line); the brown line represents Tair = 0°C. (b) $Tb_{V36A}$ (dark blue line), $Tb_{V36D}$ (dark green line), DAV36 (light green line) and $DAV36_{ice}$ (red line); the black line represents DAV=10 K. The inset map in (b) illustrates the annual mean SIC and the route of the buoy from multi-year ice to first-year ice. The black arrows indicate the cases that melt events were recognized by $DAV36_{ice}$ rather than DAV36.**

[Figure]

WS-*Weddell Sea*    IO-*Indian Ocean*    PO-*Pacific Ocean*    RS-*Ross Sea*

BAS-*Bellingshausen Amundsen Sea*    AIS-*Antarctic ice sheet*

**Figure 3. Map of the different regions across the pan-Antarctic.**

[Figure]

**Figure 4.** Annual mean EMO, CMO, melting days and MDF derived by AMSR-E/2 (a-d) and ERA-Interim (e-h), also shown are the differences (AMSR-E/2 minus ERA-Interim) between the two observations (i-l).

[Figure]

**Figure 5. Normalized histograms of annual mean EMO, CMO, melting days and MDF for pan-Antarctic and different regions.**

[Figure]

**Figure 6. Daily mean Antarctic sea ice MEF and AIS MEF, the corresponding shadows indicate daily maximums and minimums.**

[Figure]

**Figure 7. Trends in (a) CMO, (b) melting days and (c) MDF, black points indicate the pixels with trends above the 90% confidence level.**

[Figure]

**Figure 8. Linkage between pan-Antarctic snowmelt and summer SAM. (a) Correlation coefficient between MDF and summer SAM, black points indicate the trends above the 90% confidence level. (b) Comparison of normalized summer SAM (cyan line), normalized annual mean Antarctic sea ice MEF (blue line) and AIS MEF (red line).**

[Figure]

**Figure 9. Annual mean CMO derived from (a) AMSR-E and (b) W09 from 2002 to 2008.**

[Figure]

**Figure 10. Comparison of AIS melt extent derived by AMSR-E/2, ERA-Interim, and PF06 from 2002 to 2017. (a) Scatter plot of daily melt extent, blue circles indicate AMSR-E/2 vs. ERA-Interim and green circles indicate AMSR-E/2 vs. PF06. (b) Daily mean melt extent derived by AMSR-E/2 (red line), ERA-Interim (blue line) and PF06 (green line), grey shadow indicates the daily maximum and minimum melt extent detected by AMSR-E/2.**

[Figure]

**Figure 11. Annual mean EMO (blue dots), CMO (black dots), melting days (cyan dots) and MDF (red dots) with the threshold for AMSR-E/2 DAV36 varying from 6-14 K.**

---

## Author Response (AR3)

We are grateful to Dr. Stef Lhermitte for his time and attention dedicated to this manuscript and for the valuable comments. Below, we answer point-by-point all the comments. The comments are reproduced in blue and the authors' responses (**AR**) are provided in black. The underlined texts are our corresponding changes in the revised manuscript. Pages (**P**) and lines (**L**) referred to in the response are directed to the revised manuscript with changes highlighted.

The main changes in the revised manuscript include:

- The manuscript was thoroughly revised and edited by native English speaking editors.
- Uncertainties and limitations of the DAV method were discussed in more detail.

**Major remarks:**

Separate comments:

1. Please have your text revised by a native English speaker. The quality of the written English can still be strongly improved.

**AR1:** We are sorry for the grammatical problems and the inconvenience they caused in reading. The manuscript was thoroughly revised and edited by native English speaking editors at American Journal Experts (AJE) (Fig. R1). We hope it now can meet the journal's standard.

[Figure]

Fig. R1 Editing certificate from the American Journal Experts (AJE).

2. In your revised version you do not discuss the effect of the DAV melt uncertainty on the later interpretations of the melt. What would be the effect of accuracy/Kappa's of 0.82 and 0.60 (for DAV_ice) on the later interpretations? Please include this uncertainty and discuss this in more detail in your revised document.

**AR2:** Yes, uncertainties and limitations exist in the DAV method. We have included this in the

Methodology Section:

> We acknowledge that ice disintegrates, brine and flooding effects in the marginal sea ice zone may play an important role in Tb variations and obscure the DAVs (Smith, 1998; Willmes et al., 2009). The DAV method may fail to work when liquid water does not refreeze or when the snowpack is still melting during heavy melt seasons. The detection of snowmelt is sensitive to the threshold used in

the DAV method. A lower (higher) threshold will result in generally more (less) satellite-based melt signals. (**P8L1-5**)

, the comparison with other methods:

Daily Tb variation is limited when liquid water does not refreeze or snowpack is still melting in warm nights during heavy melt seasons (Willmes et al., 2009; Semmens and Ramage, 2014). As a result, ERA-Interim recognized more melt events for the regions where heavy snowmelt was found and the DAV method fails to work, such as the WS, BAS, RS and the Larsen C ice shelf. This might also be the reason that the melt onset was very early in the RS while few melt events were detected by AMSR-E/2. This is also one of the reasons that AMSR-E/2 melt signals do not strictly agree with AWS positive Tair and is considered the key limitation of the DAV method. (**P11L29-31 & P12L1-4**)

The DAV method is unaffected by snow metamorphism but may fail to work when liquid water does not refreeze or snowpack is still melting during warm nights (Willmes et al., 2009). Fortunately, the Antarctic surface snowmelt is always short-lived and can refreeze quasi-instantaneously (Zheng et al., 2018; van den Broeke et al., 2010a). (**P12L25-27**)

and the Discussion Section (5.2 Uncertainties)

It should be noted that the presence of snow liquid water detected by the AMSR-E/2 DAV method does not necessarily mean that the melt energy is positive because it takes time for meltwater to refreeze. In addition, after the refreezing of surface snow, subsurface liquid water can still be detected by a radiometer due to the penetrating capacity of microwaves (Ashcraft and Long, 2006). We detect snowmelt with AMSR-E/2, while using a SIC product generated by different sensors (SMMR and SSM/I) that observe the pan-Antarctic regions at different acquisition times. The DAV36$_{ice}$ algorithm for sea ice snowmelt detection assumes that the SIC of the two passes remains unchanged, which may not be true and lead to misidentifications of melt signals due to quick sea ice drift and disintegration. The algorithm can be further improved if the twice-daily AMSR-E/2 SIC product is available in the future. Although AMSR-E/2 observed the pan-Antarctic region at the appropriate times for snowmelt detection, they may underestimate snowmelt because snowmelt can occur at any time and refreezing can be quasi-instantaneous. Other sources of the microwave remote sensing data set, such as scatterometers and synthetic aperture radar, are expected to enrich daily observations in future works. (**P13L5-15**)

**PDF Comments:**

3. P1L11. Here → In this study.

**AR3:** Done.

4. P1L11. investigated → investigate.

**AR4:** Done.

5. P1L12. delete "the".

**AR5:** Done.

6. P1L14. delete "us".

**AR6:** Done.

7. P1L15. delete "the".

**AR7:** Done.

8. P1L15. delete "in this study".

**AR8:** Done.

9. P1L17. because → as.

**AR9:** Done.

10. P1L20-21.

What average? Spatial, temporal? Rephrase!

Is it? When looking at the maps of Fig.4 it often starts in Dec and later so I don't think it makes sense to have a spatial average and you should be more specific.

Again: I don't think spatial "averages" (?) have any meaning here.

**AR10:** Mean values at every pixel with continuous records are firstly calculated over the study period, and the spatial averages are afterwards computed. We agree that it does not make sense to have a spatial average. We rephrased this sentence with a specific description of the spatial distribution pattern of melt onset:

> Continuous melt onset (CMO) ranged from August in the marginal sea ice zone to January in the Antarctic inland, and the early transient melt events occurred several days to more than two months earlier. (**P1L23-25**)

11. P1L22. What is very likely?

**AR11:** We added the correlation coefficient and p-value to clarify:

> The decreased AIS melt extent was very likely linked ($R = -0.82$, $p < 0.01$) with the enhanced summer SAM. (**P1L28-29**)

12. P2L23. I found this confusion: dry snow is by definition not melting so this is contradicting. The role of the daily freeze-thaw cycle should be therefore explained more upfront.

**AR12:** We rephrased this sentence to avoid confusion:

> The beginning of a melt season is characterized by a sharp increase in Tb due to the freeze-thaw cycle. The snow grain size increases after the liquid water refreezes. As a result, Tb may decrease during the melt season due to the increase in volume scattering (Markus et al., 2009). (**P3L6-8**)

13. P2L27. and showed that the DAV method is unaffected ...

**AR13:** Done.

14. P2L28. This techique → the DAV technique.

**AR14:** Done.

15. P2L34. DAV → and where DAV.

**AR15:** Done.

16. P3L13. I think this (+ daily freeze-thaw ) needs to be stressed earlier already in the introduction.

**AR16:** Good suggestion. This part was moved to the second paragraph in the Introduction.

17. P3L18. typo _

**AR17:** Corrected.

18. P3L25. over the Antarctic sea ice and Ice Sheet.

**AR18:** Corrected.

19. P3L25. what kind of uncertainties

**AR19:** Clarified.

> Previous studies have separately investigated surface snowmelt over the Antarctic sea ice and Ice Sheet, which may result in uncertainties in this integrated study due to the inconsistent criteria used in melt signal determination. (**P4L10-12**)

20. P3L25. this integrated study

**AR20:** Corrected.

21. P3L27. Does the pan-Antarctic exist as a term? Perhaps rephrase to the pan-Antarctic region throughout the text.

**AR21:** Good suggestion. "pan-Antarctic" was changed to "pan-Antarctic region" throughout the text.

22. P4L13. What would be the efffect of timing difference between SIC from NSIDC-0079 from SMMR/SMMI and Tb from AMSRE? Please clarify

**AR22:** We discussed this issue in Section 5.2 (Uncertainties)

We detect snowmelt with AMSR-E/2, while using a SIC product generated by different sensors (SMMR and SSM/I) that observe the pan-Antarctic regions at different acquisition times. The DAV36$_{ice}$ algorithm for sea ice snowmelt detection assumes that the SIC of the two passes remains unchanged, which may not be true and lead to misidentifications of melt signals due to quick sea ice drift and disintegration. The algorithm can be further improved if the twice-daily AMSR-E/2 SIC product is available in the future. (**P13L8-12**)

23. P4L14. I think it is important to indicate which satellite data are used in this product to avoid clarity with reference to AMSRE/2

**AR23:** Good suggestion. More details about the SIC product were added:

The Bootstrap daily sea ice concentration (SIC) product generated by the SMMR and SSM/I Tb time series was used to mask sea ice in this study (Comiso, 2017). (**P5L2-3**)

24. P4L15. pixels with SIC greater than 80% for less than 5 days were not included in the analyses →
only pixels with more than 5 days of SIC>80% over the 2002-2017 period were analysed

**AR24:** Done.

25. P4L18. The ERA-Interim reanalysis includes → that includes

**AR25:** Done.

26. P5L5. I think there is some confusion here between what you define as melt and these single channels methods mostly fail over (daily) refrozen snowpacks. The difference between freeze-thaw and it's daily / seasonal effects on Tb need to be discussed more clearly.

**AR26:** Melt is defined as the presence of snow liquid water both in the DAV method and the single-channel methods. We have explained this issue in the revised manuscript:

Therefore, snowmelt can be detected via microwave radiometry by identifying the sharp changes in microwave measurements caused by the presence of snow liquid water (Serreze et al., 1993; Liu et al., 2005). (**P2L25-27**)

The definition of freeze-thaw and it's daily/seasonal effects on Tb were discussed more clearly:

Tb varies greatly with the appearance and disappearance of snow liquid water (i.e., freeze-thaw cycle). The evolution of Tb with increasing liquid water can be divided into energy saturation and energy dampening phases. The initial increases in Tb are accompanied by the amplification of $\varepsilon$ until the liquid water reaches a certain level. The subsequent energy dampening phase is characterized by monotonic decreases in Tb, which are caused by the increase in snow-air interface reflectivity due to the amplified real part of the refractive index (Kang et al., 2014). Previous studies have emphasized the process by which Tb increases when snowpack starts to melt. However, Tb decreases after reaching a peak when the volumetric liquid water is approximately one percent (Kang et al., 2014). The decline in Tb can also be caused by enhanced volume scattering as the snow grain size increases during the melt seasons. Single-channel methods may fail to work when the Tb of a melting snowpack is even lower than the winter mean in the late melt season due to energy dampening and snow metamorphism. Diurnal freeze-thaw cycles are prevalent in polar regions (Hall et al., 2009; Willmes et al., 2009; van den Broeke et al., 2010b). DAV shows very little variation in frozen seasons, but is very sensitive to the diurnal freeze-thaw cycles throughout the melt seasons (Zheng et al., 2018). (**P5L28-31 & P6L1-8**)

27. P5L5. I guess this is only the case in refrozen state?.

**AR27:** This phenomenon can also be found in melting state. Figure R2 shows an example at the Zhongshan Station (69.37°S, 76.38°E) (Zheng et al., 2018). Tb decreases gradually in the late melt season, and can be even lower than the winter mean due to the influence of snow metamorphism and energy dampening. We have discussed this issue in the revised manuscript (Please see **AR26** for full details).

[Figure]

Figure R2. The comparisons of ascending Tb (blue lines), descending Tb (green lines), and DAV (red lines) of each AMSR-E band at the Zhongshan Station from 1 July 2010 to 30 June 2011 (adopted from Zheng et al., 2018). (a) and (b) are the horizontal and vertical observations, respectively. Tair below and above 0 °C is indicated using a yellow line and magenta dots in the bottom panel; the black lines represent Tair = 0 ℃; grey shadows indicate the melt season interpret from positive Tair.

28. P6L2. much lower Tb → much lower Tb's / a much lower Tb.
**AR28:** Done.

29. P6L23-24. This reference is missing from the references.

**AR29:** Fleiss et al. (2003) was included in the reference list:

> Fleiss, J. L., Levin, B. and Paik, M. C.: Statistical Methods for Rates and Proportions, Third Edit., Wiley, Hoboken, New Jersey. pp.604, 2003. (**P15L21-22**)

30. P7L28. I guess you mean: least squares method.

**AR30:** Corrected.

31. P8L9-12. This should be part of your study area.

**AR31:** This part was moved to Section 2 (Data sets and study area).

32. P8L13. What is the average? An average of what?

**AR32:** Mean values at every pixel with continuous records are firstly calculated over the study period, and the spatial averages are afterwards computed. We agree that it does not make sense to have a spatial average over the pan-Antarctic region. We rephrased this sentence:

> The CMO ranges from August in the marginal sea ice zone to January in the Antarctic inland, and the pan-Antarctic region EMO arrived 53 days earlier on average (Table 1). (**P9L18-21**)

33. P8L17-20. I think this is a bit of a strange argument. If there is open water, it seems obvious it cannot melt.

**AR33:** We removed this part in the revised manuscript.

34. P9L6. what doe you mean by concentrate? peaked?

**AR34:** "concentrated" was changed to "peaked".

35. P11L27-28. This is contradicting with your definition of melt: the presence of water in the snowpack. You should be much more clear on the exact definitions here and I guess you mean negative energy balance?

**AR35:** Good catch. We revised this sentence to clarify:

[revised manuscript text omitted]

---

## Author Response (AR4)

Dear Dr. Stef Lhermitte,

Thank you so much for the suggestion.

Fig. 7 and Fig. 8 were redrawn to make the significant regions clear.

We greatly appreciate all your help and that of the referees concerning the improvement of this paper.

Sincerely,

Lei Zheng on behalf of all the authors

Comment:

Before final publication, however, I would like to ask you to check if you can make the significant regions in Fig 7+8 more clear to the reader? Currently the black points are difficult to see and seem to correspond to dark blue/red (except when zooming in a lot).

Fig. 7 and Fig. 8 were redrawn to make the significant regions clear.

[revised manuscript text omitted]

**WS**-*Weddell Sea*      **IO**-*Indian Ocean*      **PO**-*Pacific Ocean*      **RS**-*Ross Sea*

**BAS**-*Bellingshausen Amundsen Sea*      **AIS**-*Antarctic ice sheet*

**Figure 1. Map of the different regions across the pan-Antarctic region.**

[Figure]

**Figure 2. The comparison between Tair and satellite observations (AMSR-E from 2002 to 2011 and AMSR2 from 2012 to 2017) at Zhongshan Station, including daily maximum Tair (purple line), Tb$_{V36A}$ (dark blue line), Tb$_{V36D}$ (dark green line) and DAV36 (red line); Brown and black lines represent Tair = 0℃ and DAV = 10 K.**

[Figure]

**Figure 3. Meteorological and satellite measurements along a sea ice buoy in the Weddell Sea from 1 September, 2014 to 1 May, 2015. (a) Snow depth (cyan line), daily maximum Tair (pink line), and SIC (olive line); the brown line represents Tair = 0°C. (b) $Tb_{V36A}$ (dark blue line), $Tb_{V36D}$ (dark green line), DAV36 (light green line) and $DAV36_{ice}$ (red line); the black line represents DAV=10 K. The inset map in (b) illustrates the annual mean SIC and the route of the buoy from multi-year ice to first-year ice. The black arrows indicate the cases that melt events were recognized by $DAV36_{ice}$ rather than DAV36.**

[Figure]

**Figure 4. Annual mean EMO, CMO, melting days and MDF derived by AMSR-E/2 (a-d) and ERA-Interim (e-h), also shown are the differences (AMSR-E/2 minus ERA-Interim) between the two observations (i-l).**

[Figure]

**Figure 5. Normalized histograms of annual mean EMO, CMO, melting days and MDF for pan-Antarctic region and different regions.**

[Figure]

**Figure 6. Daily mean Antarctic sea ice MEF and AIS MEF, the corresponding shadows indicate daily maximums and minimums.**

[Figure]

[Figure]

[Figure]

**Figure 7. Trends in (a) CMO, (b) melting days and (c) MDF, black points indicate the pixels with trends above the 90% confidence level.**

[Figure]

[Figure]

**Figure 8. Linkage between pan-Antarctic region snowmelt and summer SAM. (a) Correlation coefficient between MDF and summer SAM, black points indicate the trends above the 90% confidence level. (b) Comparison of normalized summer SAM (cyan line), normalized annual mean Antarctic sea ice MEF (blue line) and AIS MEF (red line).**

[Figure]

**Figure 9. Annual mean CMO derived from (a) AMSR-E and (b) W09 from 2002 to 2008.**

[Figure]

**Figure 10. Comparison of AIS melt extent derived by AMSR-E/2, ERA-Interim, and PF06 from 2002 to 2017. (a) Scatter plot of daily melt extent, blue circles indicate AMSR-E/2 vs. ERA-Interim and green circles indicate AMSR-E/2 vs. PF06. (b) Daily mean melt extent derived by AMSR-E/2 (red line), ERA-Interim (blue line) and PF06 (green line), grey shadow indicates the daily maximum and minimum melt extent detected by AMSR-E/2.**

[Figure]

**Figure 11. Annual mean EMO (blue dots), CMO (black dots), melting days (cyan dots) and MDF (red dots) with the threshold for AMSR-E/2 DAV36 varying from 6-14 K.**